# Sarcopenia as a Multisystem Disorder—Connections with Neural and Cardiovascular Systems—A Related PRISMA Systematic Literature Review

**DOI:** 10.3390/life16010068

**Published:** 2026-01-01

**Authors:** Cristina Popescu, Sorina-Maria Aurelian, Andrada Mirea, Constantin Munteanu, Andreea-Iulia Vlădulescu-Trandafir, Aurelian Anghelescu, Corina Oancea, Ioana Andone, Aura Spînu, Andreea-Valentina Suciu, Simona-Isabelle Stoica, Sandra-Monica Gîdei, Valeria-Mădălina Alecu, Costina-Daniela Gîță, Nadina-Liana Pop, Vlad Ciobanu, Gelu Onose

**Affiliations:** 1Faculty of Medicine, University of Medicine and Pharmacy “Carol Davila”, 020022 Bucharest, Romania; cristina_popescu_recuperare@yahoo.com (C.P.); stoica.simona@umfcd.ro (S.-M.A.); andrada.mirea@gmail.com (A.M.); corina.oancea@umfcd.ro (C.O.); ioana.andone@umfcd.ro (I.A.); aura.spinu@umfcd.ro (A.S.); andreea-valentina.spiroiu@drd.umfcd.ro (A.-V.S.); sandra-monica.gidei@drd.umfcd.ro (S.-M.G.); costina-daniela.gita@drd.umfcd.ro (C.-D.G.); gelu.onose@umfcd.ro (G.O.); 2Neuromuscular Rehabilitation Clinic Division, Teaching Emergency Hospital “Bagdasar-Arseni”, 041915 Bucharest, Romania; aurelian.anghelescu@umfcd.ro (A.A.); simona-isabelle.stoica@umfcd.ro (S.-I.S.); 3Gerontology and Geriatrics Clinic Division, St. Luca Hospital for Chronic Illnesses, 041915 Bucharest, Romania; valeria-madalina.mihai@drd.umfcd.ro; 4National Clinical Centre of Neurorehabilitation for Children “Dr. N. Robanescu-Pădure”, 041408 Bucharest, Romania; 5Faculty of Medical Bioengineering, University of Medicine and Pharmacy “Grigore T. Popa” Iasi, 700454 Iași, Romania; 6Faculty of Midwifery and Nursing, University of Medicine and Pharmacy “Carol Davila”, 020022 Bucharest, Romania; 7The National Institute for Medical Assessment and Work Capacity Rehabilitation, 050659 Bucharest, Romania; 8Rehabilitation Clinic Division, Carol Davila Nephrology Hospital, 020021 Bucharest, Romania; 9Department of Physiology, Iuliu Hațieganu University of Medicine and Pharmacy, 400006 Cluj-Napoca, Romania; popnadina@yahoo.com; 10“Dr. Constantin Papilian” Military Emergency Hospital, 400132 Cluj-Napoca, Romania; 11Computer Science Department, Politehnica University of Bucharest, 060042 Bucharest, Romania; vlad.ciobanu@upb.ro

**Keywords:** sarcopenia, muscle strength, rehabilitation, physical performance, neuromuscular junctions, mitochondrial dysfunction, inflammation, aging

## Abstract

Background: Sarcopenia, which has traditionally been considered to be an exclusively geriatric syndrome, has an increased frequency within the general population and fosters interest in its complex neuromuscular, cardiovascular, and metabolic basis. The current systematic review, adopting the recognized Preferred Reporting Items for Systematic Reviews (PRISMA) methodology, seeks to highlight current evidence on the underlying mechanisms as well as approaches to sarcopenia diagnosis and management. Methods: A comprehensive search of major international databases identified studies published between January 2023 and December 2024, from which 42 articles were retained according to prespecified criteria. To further enrich the present work, eleven additional studies of high relevance were included. Results: The selected literature describes sarcopenia’s multifactorial pathophysiology, including mitochondrial dysfunction, neuromuscular junction (NMJ) degeneration, chronic inflammation, anabolic resistance, endocrine and metabolic dysregulation, altered motor-unit remodeling, and molecular alterations. Diagnostic methods focus on functional assessments, especially muscle strength and physical performance. In addition, imaging techniques and new circulating biomarkers enhance precision in specific situations. Over the years, rehabilitation has proven to be one of the most effective therapeutic approaches. Complementary strategies, ranging from nutritional optimization to pharmacologic modulation of the renin–angiotensin system, show promise in specific patient subsets. Discussion and Conclusions: As supported by the works collected within the current study, future approaches will need to consider sarcopenia as a multifactorial disease that goes beyond aging.

## 1. Introduction

Sarcopenia (from the Greek “sarx” = flesh and “penia” = loss) is recognized as one of the most important geriatric syndromes and is characterized by a progressive loss of skeletal muscle mass and function, leading to declines in strength, physical performance, and mobility (Figure 1). It is strongly associated with frailty, disability, an increased frequency of falls, and fragility fractures. These consequences contribute to a reduced capacity of performing activities of daily living (ADL), consequently: personal autonomy, elevated levels of psychological distress—all resulting in reduced quality of life (QoL)—and further possible augmented length of hospitalization, cognitive impairment, decline in cardiovascular health and increased mortality [1,2,3,4,5,6,7]. Initially, it was defined mostly by muscle mass reduction; however, it is now viewed as a broader syndrome affecting muscle strength and physical performance. As a result, it has been formally acknowledged and included in the International Classification of Diseases, Tenth Revision (ICD-10) under the code M62.84 [2,8,9,10]. Major consensus groups, including the European Working Group on Sarcopenia in Older People 2 (EWGSOP2) and the Asian Working Group for Sarcopenia (AWGS), now prioritize muscle strength—rather than muscle quantity—as the most reliable predictor of clinical evolution [2,9,11,12,13].

Representing about 40% of the body’s mass, skeletal muscles are known as the main organ of movement. But the narrative is broader, as the musculoskeletal system also acts as an active endocrine interface due to its capacity to produce a diverse spectrum of myokines that regulate metabolic functions at the organism level [14,15]. Sarcopenia is also associated with cognitive frailty, suggesting shared mechanisms involving neurodegeneration, chronic inflammation, and physical inactivity, and with non-alcoholic fatty liver disease (NAFLD), where metabolic changes lead to muscle breakdown [16,17].

Sarcopenia usually manifests from the sixth decade of life onward and progresses at a rate of approximately 0.5–2% muscle loss per year [4,8,18]. The prevalence of this condition in individuals under 60 years of age has been estimated to range between 8% and 36%. These incongruences occur not only as a result of variation in the conceptual framework of diagnosis and methods of measurement, but also from the varying populations being studied, thereby highlighting the challenges of developing a definition that can be generalized [2,3,8,12,19,20,21,22]. Sarcopenia is therefore considered a key biological substrate of frailty and a major determinant of late-life disability, specifically marked impairment of gait speed, grip strength, stair-climbing ability, and overall functional autonomy [3,5,16].

Apart from the main risk factor—aging—there are some known contributors to the development of this condition. These are malnutrition, physical inactivity, inflammation, hormonal changes, oxidative stress, circadian rhythm dysregulation, vitamin D deficiency, and multimorbidity. So, sarcopenia is no longer regarded as a condition exclusively affecting the elderly; it may also occur in younger individuals with chronic illnesses and is thus classified as either primary (age-related) or secondary (disease-related) [4,9,13,19,23,24].

At the cellular level, sarcopenia results from complex biological changes involving mitochondrial dysfunction and dysregulation of protein metabolism, with interrelated mechanisms that are not fully understood [14,25]. Due to its multifactorial causes and significant functional consequences, and in the context of accelerated demographic aging, globally, sarcopenia has come to be recognized as a pressing public health issue requiring early detection and preventive interventions [4,12,23,26,27].

Hence, the priority of such strategies is underscored by the global demographic transition that estimates, by 2030, one out of every six persons in the global population will be ≥60 years. This number is expected to exceed 2.1 billion by 2050, with a projection to see thrice as many persons above 80 years, as reported by the World Health Organization (WHO) [21,28]. As a brief related parenthesis, in the last two decades, the definition of elderly has referred to a person aged 65 years or older [29].

This contemporary change underscores the importance not only of extending survival but also of placing value on those additional years, shifting healthcare priorities from longevity itself toward healthy, active aging [18,26,27].

Sarcopenia is now considered a potentially treatable and thus reversible condition, especially in the early stages. People in the preliminary or presarcopenic phase may especially benefit from early treatment, which can prevent and thus reverse musculoskeletal progression. Prospective findings in the study show that the course of the disease is not one-dimensional; in fact, in the course of the five-year tracking, the proportion of subjects in the probable group either developing the syndrome in full or reverting to normal was not different (10.7% and 10.3%, respectively). Several treatment modalities—physical therapy, dietary interventions, and the use of pharmacologic therapies—are currently being investigated for the possible restoration or at least the conservation of the integrity of skeletal muscles in sarcopenia [7].

Yet, despite the growing body of research, important uncertainties persist regarding the underlying mechanisms, optimal diagnostic thresholds, and the most effective strategies for prevention, treatment, and rehabilitation. The unrelenting heterogeneity across definitions and assessment tools continues to hinder clinical implementation, early detection, and comparability between studies. At the same time, the rapid demographic aging worldwide amplifies the need to advance from theoretical understanding to early identification and actionable interventions.

In this respect, a thorough and up-to-date synthesis of the existing evidence may be considered a relevant task. The objective of this PRISMA-based systematic review, is to provide a synthesis of the most recent evidence related to sarcopenia as a multisystem disorder, adopting a purposeful approach towards the neuromuscular, neurological, and cardiovascular aspects, and also combining the integrated and additional relevant areas that impact upon the phenomenon, such as metabolic and endocrine dysregulation, inflammation, gut and muscle interrelationships, and innovative rehabilitation techniques. We will also provide a summary of the current diagnostic practices (see Figure 2) and discuss preventive and therapeutic–rehabilitative strategies, aiming to support clinical decision-making and to outline priorities for future research.

While this review adopts a broad and integrative perspective, its primary emphasis is placed on the biological and multisystem mechanisms of sarcopenia. Diagnostic approaches and therapeutic strategies have also been covered within this review, reflecting the current availability and heterogeneity of interventional evidence.

## 2. Materials and Methods

### 2.1. Study Design

We performed a systematic literature review following the aforementioned PRISMA methodology [30,31] to synthesize contemporary evidence on the pathophysiological mechanisms of sarcopenia and its clinical management, with a particular attention on physical exercise interventions and virtual environment- or technology-assisted rehabilitation strategies in older adults, where evidence was available (Figure 3).

This review aims to explore both the pathophysiological underpinnings, including neuromuscular, cardiovascular, and metabolic processes, as well as the treatment techniques used for preventing or reversing the functional decline associated with sarcopenia.

### 2.2. Search Strategy

A thorough electronic search was conducted for articles published from 1 January 2023, to 31 December 2024, across four major international databases: National Center for Biotechnology Information (NCBI)/PubMed (last accessed on 21 August 2025) [32], NCBI/PubMed Central (PMC—last accessed on 21 August 2025) [33], Elsevier (last accessed on 21 August 2025) [34], and the Physiotherapy Evidence Database (PEDro—last accessed on 21 August 2025) [35]. The search strategy utilized contextually specific keyword combinations/syntaxes: “sarcopenia”, “physical exercise”, “virtual reality”, “VR”, “virtual augmented reality”, “virtual environment”, “angiotensin-converting enzyme inhibitors”, “ACEIs”, “angiotensin receptor blockers (ARBs)”, “activin receptor type 2b (ACVR2B)”, “motor end plates”, “myasthenia gravis”, “Parkinson’s disease”, “Alzheimer’s disease”, “stroke”, “spinal muscular atrophy”, “diabetic polyneuropathy”, “amyotrophic lateral sclerosis”, “multiple sclerosis”, “spinal cord injury”, “titin”, “desmin”, “nebulin”, “myosin”, “sarcomere” (see Appendix B).

### 2.3. Eligibility Criteria

To ensure methodological rigor, we applied predefined eligibility criteria. We included papers that: (1) contained original research articles (clinical trials, observational studies, mechanistic studies, or interventional trials), systematic reviews or meta-analyses; (2) investigated sarcopenia in the context of pathophysiology, physical rehabilitation, or digital/virtual technology–assisted interventions (e.g., virtual reality, augmented reality, or virtual environments); (3) involved human participants, preferably older adults, but also animal-only studies highly relevant to molecular mechanisms of neuromuscular degeneration, cardiovascular and metabolic impairments; (4) were published in English and available as full text formats, open access; and (5) were published in journals indexed in International Scientific Indexing (ISI)/Web of Science [31,36]. We excluded: conference abstracts, non-English publications, papers without an accessible free full text.

Given the marked heterogeneity of study designs, populations, and outcome measures across the sarcopenia literature, this review included not only original clinical and experimental studies but also high-quality evidence syntheses. The relevance of systematic reviews and meta-analyses, particularly in those clinical and research topics in which primary literature is dispersed and methodologically heterogeneous, was considered to support an integrative interpretation of neuromuscular, neural, cardiovascular, and metabolic pathways rather than to replace the analysis of primary evidence.

### 2.4. Screening and Quality Assessment

The initial search identified a total of 389 articles (115 from Elsevier, 260 from PubMed Central, 14 from PEDro). The next step was the removal of duplicate records, and thus 223 articles remained and were further verified for publication in ISI Web of Science. Of these, 188 were confirmed as ISI-indexed and advanced to methodological quality assessment. A customized scoring system/algorithm derived from PEDro [37,38,39]—adapted to assess validity, methodological transparency, and clinical relevance—was used to evaluate scientific quality. Furthermore, the final synthesis included only studies that met a methodological threshold of ≥4 points (indicating at least fair quality). Those studies were selectively incorporated for detailed examination. The PEDro-derived score was applied uniformly across all included studies, regardless of study design. Considering the integrative and multidisciplinary nature of the review, as well as heterogeneity of the included studies, the score was not used to assess trial-specific internal validity, but rather as a pragmatic indicator of scientific visibility and minimum necessary quality criteria. This approach was chosen to allow a consistent appraisal across heterogeneous study types and to preserve methodological transparency. Both the selection of the articles in the review and the scoring of individual articles were performed in parallel by two reviewers, and any differences were resolved through discussion and consensus among two other reviewers. Complete identification of articles can be found in Appendix A. After excluding studies that did not meet eligibility criteria, 56 papers qualified for the review.

### 2.5. Final Selection of Studies

After the full-text evaluation and exclusion of articles not satisfying the pre-defined criteria, including those that were not sufficiently methodologically robust or that presented incomplete clinical relevance, a total of 42 studies were selected for the qualitative analysis (see Appendix C).

To further consolidate the scientific value of the present review, an additional set of 11 articles identified independently during the extended literature search was incorporated in the final synthesis. These studies were identified through manual reference screening and expert knowledge, particularly in areas insufficiently captured by the predefined database search strategy. Some of these articles have focused on pediatric cases of sarcopenia or have considered additional pathophysiological mechanisms; others were published within the timeframe following the end of the predefined search window for the review and were included to ensure that the review reflects the most recent evidence in rapidly evolving fields.

Figure 1, as indicated in the PRISMA guidelines, presents a flow diagram for the selection process. The systematic review was prospectively registered in the International Prospective Register of Systematic Reviews (PROSPERO) Database under the number CRD420251132900.

## 3. Results

### 3.1. Muscle Structure and Physiopathological Mechanisms

Sarcopenia appears to be associated with a convergence of interrelated biological processes (Figure 4)—including mitochondrial impairment, reduced muscle regeneration capacity, chronic low-grade inflammation, progressive neuronal loss, and a decline in cellular systems that maintain protein homeostasis, as well as endocrine and metabolic dysregulation—which are collectively linked to progressive impairment of neuromuscular integrity and functional capacity in aging individuals and, to varying degrees, to alterations in cardiometabolic and/or vascular states [14,40]. Skeletal muscle is not a static tissue but rather a metabolically active, continuously remodeled organ, maintained by a finely regulated balance between protein synthesis and degradation. This equilibrium is influenced by nutritional intake, physical activity, and catabolic conditions such as fasting, and may be disrupted by insulin resistance, systemic inflammation, prolonged corticosteroid exposure, chronic kidney disease (and especially dialysis), and/or age- and disease-related reductions in anabolic hormones. As these influences accumulate, the skeletal muscle becomes progressively refractory to anabolic stimuli—a phenomenon known as anabolic resistance—in which protein synthesis no longer adequately responds to resistance training (RT) or essential amino acid intake [2,7,16,22,41].

#### 3.1.1. Mitochondrial Dysfunction and Metabolic Stress

Impaired mitochondrial health, including damaged mitochondrial DNA, has been reported across multiple organ systems but appears particularly relevant in tissues with high energy demands—namely, the heart, central and peripheral nervous systems, and the skeletal and smooth muscles—where it has been associated with adverse structural and functional outcomes, such as sudden cardiac arrest, accelerated neurodegeneration, and various myopathic disorders. When mitochondrial compromise surpasses a certain biochemical threshold, ATP depletion together with impaired coupling efficiency initiates apoptotic pathways and destabilizes cellular survival mechanisms [14,42]. This does not happen in isolation, as with aging, mitochondrial alterations are coupled with adverse modifications to the cellular redox environment, with escalating ROS levels coupled with reduced endogenous protection, including reduced glutathione, ascorbic acid, and other protective mechanisms. As the availability of antioxidants—including vitamins C and E, lipoic acid, and the glutathione-linked enzymatic machinery—continues to fall, cells accumulate misfolded or damaged proteins, and compensatory processes such as autophagy and mitophagy become less effective. This persistent disequilibrium between oxidant pressure and antioxidant defenses fosters incremental molecular injury and hastens the progression of sarcopenia [14,22,42].

In skeletal muscle, age-related mitochondrial dysfunction has been consistently described in association with sarcopenia, and is characterized by impaired oxidative phosphorylation, excessive production of ROS, and inadequate clearance of damaged proteins and organelles due to the breakdown of mitochondrial and cytosolic proteostasis. Gene-expression analyses have shown that older muscle exhibits reduced mitochondrial translational activity, a defect that mirrors the phenotype of young mice lacking peroxisome proliferator-activated receptor gamma coactivator-1α (PGC-1α)—a master regulator of mitochondrial biogenesis and metabolic adaptation. Experimental studies indicate that re-expression of PGC-1α can reinduce the synthesis of mitochondrial ribosomal proteins in an estrogen receptor-alpha–dependent manner, highlighting its regulatory role in mitochondrial adaptation [25]. However, the ability of muscle tissue to mount an appropriate mitochondrial unfolded protein response (UPRmt) becomes progressively blunted with age, while the accumulation of ubiquitinated proteins reflects reduced function in cell survival signaling mechanisms. Researchers continue to investigate several circulating mediators—especially myostatin (a TGF-β–related molecule) and IGF-1—due to their potential to indicate early disturbances in mitochondrial function or broader defects in protein regulation [14,15,40]. Moreover, another study shows that chronic activation of cyclooxygenase-2 (COX-2) and toll-like receptor 2 signaling can worsen oxidative damage. With time, all the above-mentioned molecular shifts become visible as muscles lose both size and strength. Individual fibers shrink, and the normal cycle of protein renewal is blunted. When oxidative stress and inflammation persist, they fuel further breakdown, supporting an association between mitochondrial dysfunction and the declining physical function [14].

The overall effects of mitochondrial impairment in sarcopenia mirror patterns seen in both cardiovascular and neurodegenerative ailments. Specific to this concept, it is worth mentioning that the elderly have lower Nicotinamide Adenine Dinucleotide (NAD+) levels and inadequate sirtuin (SIRT) signaling, all contributing to reduced mitochondrial stability and further may precipitate endothelial dysfunction, rhythm instability, and ultimately the emergence of HF. Moreover, impairment of the PGC-1α/peroxisome proliferator-activated receptor (PPAR) signaling axis adds an additional layer of metabolic stress, reducing mitochondrial biogenesis and altering energy balance not only in the myocardium but also in skeletal muscle and the nervous system [42]. In this context, several studies have explored therapeutic strategies that help preserve mitochondrial integrity or limit oxidative damage, providing a biological rationale for further investigation in the setting of sarcopenia [9,14,20].

#### 3.1.2. Proteostatic Imbalance, Impaired Regeneration and Anabolic Resistance

Beyond problems with energy metabolism, aging muscle also loses its balance in protein regulation. The rate of new protein formation slows, while degradation through the ubiquitin–proteasome system accelerates, and the tissue becomes less responsive to anabolic signals [7,40,41]. Normally, skeletal muscle can still repair itself thanks to satellite cells (small resident stem cells that activate after injury or exercise, contribute to muscle repair and are guided by growth factors such as IGF-1) [7,15,17]. With aging and/or chronic diseases, however, both the number and the efficiency of these cells decrease, especially within fast-twitch type II fibers, resulting in incomplete repair. Insulin resistance, persistent low-grade inflammation, and the dysfunctional communication between muscle and fat further balance the scale toward net loss of protein. Over time, these processes help explain why sarcopenia so often appears alongside frailty, cognitive decline, or even cardiac dysfunction—an overlap increasingly described in the literature as cardio-sarcopenia (see also further) [7,40].

#### 3.1.3. Muscle Architecture Remodeling and Decline in Contractile Quality

As these molecular disturbances build up, the overall structure of the muscle starts to change, and the muscle’s ability to contract declines. A natural interplay of fibers exists in every muscle: slow, oxidative type I and IIa fibers that are more resistant to fatigue, and fast, glycolytic type IIb fibers that can contract powerfully but are fatigue-prone [7,15]. The proportion between these fibers varies from one muscle to another, which partly explains why some muscles waste earlier than others. Within each muscle cell, the primary contractile elements—sarcomeres—are made of actin and myosin filaments that interlace at the Z-disc and M-line. In order to maintain genomic integrity, sarcomeres contain a network of accessory proteins, such as titin, nebulin, α-actinins (ACTN2/3), myozenin (MYOZ1-3), myomesin, filamin C, and obscurin [40]. The costamere is a specialized multiprotein cluster (dystrophin–glycoprotein complexes, integrins, vinculin, ankyrin, talin, focal adhesion kinase, and desmin) that anchors the sarcomere to the sarcolemma and further connects to the extracellular matrix, functioning as a mechanoresponsive signaling interface that transmits contractile mechanical signals to mTORC1 via phosphorylation-dependent mechanotransduction [2].

Fast fibers mainly contain myosin heavy chain (MyHC)-2a, MyHC-2x, and MyHC-2b (MYH2, MYH1, MYH4) and fast myosin light chains such as MLC-1/3 (MYL1), whereas slow fibers are predominantly made of MyHC-1 (MYH7) and slow isoforms of myosin light chains, troponin, and tropomyosin. With aging, proteomic analyses reveal a gradual shift from fast- to slow-contracting fiber phenotypes—driven by the selective denervation of fast motor units and preferential collateral reinnervation of slow fibers—which favors endurance but reduces maximal power output. Several studies report that qualitative alterations in muscle architecture may precede measurable loss of muscle mass [7,40,43].

#### 3.1.4. Inflammatory and Catabolic Signaling Dominance

Sarcopenia has also been associated with persistent, low-grade inflammation and oxidative stress that act through tightly interconnected catabolic systems, accelerating proteolysis and impairing regeneration. Oxidative stress was already discussed in the context of mitochondrial dysfunction. When it comes to inflammation, both acute and chronic forms increase the levels of pro-inflammatory cytokines (e.g., TNF-α, IL-1, IL-6, TGF-β) that suppress mTORC1 activity and activate NF-κB and forkhead box O (FOXO), both catabolic axes, inducing muscle atrophy genes such as FBXO32 and TRIM 63 that encodes the protein named MuRF1. Together, these alterations are associated with a sustained predominance of catabolic over anabolic signaling [2,7,12,14,22,44,45]. It is important to note that an important difference in the inflammatory response is observed between men and women, with implications for sarcopenia [23].

These cytokines simultaneously stimulate proteolytic pathways: ubiquitin–proteasome, autophagic–lysosomal, calpain, and caspase cascades, all of which degrade structural and contractile proteins. Under physiological conditions, moderate ROS levels act as signaling molecules, but their excessive accumulation disturbs mitochondrial metabolism, promotes fibrosis, fatty infiltration, protein degradation and accelerates apoptosis. COX-2 inhibition, interestingly, has shown promise in attenuating inflammation-induced fibrosis [14,44].

At the molecular level, inflammatory cytokines, disruption of circadian rhythm, and insulin resistance converge to impair anabolic signaling through the IGF–1-PI3K–AKT–mTORC1 pathway. Attenuation of this pathway has been associated with reduced protein synthesis, while the concomitant reduction in mTORC1 and AKT activity during immobilization or disease further curtails myocellular growth [12,13,14]. By contrast, sustained activation of mTORC1 orchestrates protein and lipid anabolism, thereby supporting cellular expansion; its suppression under inflammatory conditions has been linked to molecular patterns consistent with sarcopenic decline [22].

Superimposed upon these alterations are molecular hallmarks of aging. The gradual accrual of advanced glycation end-products (AGEs)—formed by non-enzymatic glycation between reducing sugars and proteins with a slow turnover—perpetuates oxidative stress, inflammation, stiffening of the extracellular matrix, and promotes a pathogenic vicious cycle. Consistent with this, clinical findings corroborate these mechanistic insights, demonstrating inverse correlations between circulating AGE markers and indicators of muscle mass and strength and offering promise for an early diagnosis of sarcopenia [6].

#### 3.1.5. Adipose–Muscle–Bone Crosstalk and the Metabolic Shift Toward Sarcopenic Obesity

Discussing inflammation in more detail, studies have shown that chronic low-grade inflammation within the adipose tissue—driven by M1 macrophages, immunosenescent T cells, and cells with SASP—promotes systemic insulin resistance and oxidative stress [4,46]. In parallel, ectopic fat infiltrates skeletal muscle, inducing lipotoxicity, mitochondrial dysfunction, impaired contractility, and local cytokine release. The muscle itself then behaves as a dysregulated endocrine organ, overproducing catabolic myokines, such as myostatin (one of the most important regulators of muscle mass), TNF-α, IL-6, and IL-8. This bidirectional inflammatory interplay has been proposed as a biological basis for sarcopenic obesity, where muscle weakness and qualitative deterioration may precede loss of muscle mass [4,15,16,23,47]. The definitions and quantification of sarcopenic obesity have very high heterogeneity between studies and are associated with chronic diseases such as non-alcoholic fatty liver disease, liver fibrosis, cirrhosis, and atherosclerotic cardiovascular disease [47].

Another important discussion is about myosteatosis, described as a pathologic derangement of muscle architecture, by the insidious intramuscular infiltration of adipose tissue. This accompanies several mechanisms involved in sarcopenia. On CT imaging, this process is identified as a reduction in muscle radiodensity—a surrogate of compromised tissue quality rather than mere mass loss. As individuals age, this pattern becomes increasingly prevalent and aligns closely with the broader clinical picture of sarcopenia, frailty, and higher risk of falls [7,14,47]. In addition, this adipose–muscle intersection is influenced by hormonal factors (reduced 17β-estradiol levels contribute to sarcopenia) and neurotrophic constituents [brain-derived neurotrophic factor (BDNF) supports neuroplasticity, muscle regeneration and its overall health, and it is induced by physical activity]. Not least, sex and circadian rhythm also influence this axis. In women, diminished IGF-1 signaling together with poor nutritional status increases susceptibility to insulin resistance and sarcopenia. An inverse relationship between muscle strength and circulating levels of myostatin, follistatin, and vitamin D has been reported only in female cohorts, which is interestingly pointing toward sex-specific pathways in muscle aging and systemic adaptability [23].

Osteosarcopenia, a disorder characterized by the concomitant loss of bone and muscle (mass and/or function), illustrates the close metabolic interdependence of these tissues. Bone remodeling is influenced by muscle activity and adipokines such as leptin, adiponectin, resistin, and adipsin. When adiposity exceeds a threshold and becomes excessive, leptin resistance and the pro-inflammatory cytokine milieu disturb bone turnover and can accelerate the onset of sarcopenic obesity. In this setting, mitochondrial dysfunction, oxidative stress, and impaired insulin signaling converge, forming a feedback loop that perpetuates metabolic dysfunction. The impact of the Coronavirus Disease 2019 (COVID-19) pandemic underscored this association, revealing numerous patients with concomitant bone and muscle loss, and emphasizing the need for early interventions targeting bone–muscle homeostasis [17,47].

#### 3.1.6. Gut–Muscle–Immune Axis and Microbial Dysbiosis

Age-related dysbiosis—which, in sarcopenia, is characterized by the loss of short-chain fatty acid (SCFA)-producing species, expansion of Proteobacteria, and increased intestinal permeability—allows translocation of lipopolysaccharide into circulation, fueling chronic low-grade inflammation, oxidative stress, and activation of catabolic adenosine monophosphate protein-kinase (AMPK)–FoxO3–Atrogin-1/MuRF1 pathways. Secondly, microbial metabolites, such as indoxyl sulfate, can cause mitochondrial toxicity. On the other hand, dysbiosis may induce cellular senescence by overproducing p16^Ink4a and SASP mediators. Some age-associated microbial species, including Bacteroides acidifaciens, have been associated with immune cell senescence, further impairing immune surveillance and promoting inflammaging. Experimental studies and early clinical trials assessing the effectiveness of physical exercise, caloric restriction, fecal microbiota transplant, and specific probiotics showed variable results regarding microbial composition and muscle outcomes [4,45]. Alterations in SCFAs production have been associated with changes in IGF-1 signaling and inflammatory status, which may be relevant to sarcopenia-related mechanisms [12].

### 3.2. Neuro-Functional Mechanisms in Sarcopenia

#### 3.2.1. Central Nervous System (CNS) Involvement

Neurological impairments have been frequently reported in association with the development and progression of sarcopenia. Worldwide, neurological diseases represent the primary cause of disability and the second-most common cause of death after cardiovascular diseases, and aging is the strongest risk factor for most neurodegenerative disorders [48]. Sarcopenia, frailty and falls are particularly prevalent in the elderly, especially with neurological disorders such as stroke, Parkinson’s disease, and dementia, but also in the oldest olds [21,49]. Regarding fall risk, medications, especially psychotropics used in the above-mentioned conditions, increase fall risk and its consequences [16]. The connection between all the aforementioned physiologically related aspects of aging and/or sicknesses affecting the CNS—i.e., brain and spinal cord—is very complex, including sarcopenia, consisting mainly of impaired neuronal function, reduced neurogenesis, chronic neuroinflammation and increased blood–brain barrier (BBB) permeability [48].

At the same time, aging microglia and a weakened BBB promote the systemic release of inflammatory mediators such as IL-6, TNF-α, tissue inhibitor of metalloproteinase 1 (TIMP-1), intercellular adhesion molecule 1, and glial fibrillary acidic protein, which closely correlate with the severity of sarcopenia and frailty [48]. At the same time, the close association between sarcopenia, cognitive decline, and impaired motor coordination highlights a bidirectional muscle–brain axis. Within the geroscience paradigm, mitochondrial dysfunction, neuroinflammation, impaired metabolic function and intracellular signaling, together with comorbidities, decreased physical activity and protein intake, are considered shared pathways associated with both sarcopenia and neurodegeneration. Sarcopenia may therefore be viewed not only as a downstream manifestation of neural aging but also as a condition potentially linked to cognitive and motor deterioration [2,12,46].

#### 3.2.2. Neuromuscular Junction and Motor Unit Degeneration

At the peripheral level, sarcopenia is strongly influenced by age-related degeneration of motor neurons and progressive instability of the NMJ. “Dying-back” axonal degeneration leads to the loss of motor units, while compensatory reinnervation by surviving neurons yields fewer but larger units—temporarily preserving mass yet diminishing fine motor control and strength [8,12].

Structural NMJ disorganization—including synaptic fragmentation, terminal regression, and reduced agrin signaling—appears early in the disease process. Experimental evidence shows that the loss of agrin and increased expression of MuSK and Lrp4 genes impair synaptic stability, whereas elevated circulating CAF and NfL levels indicate ongoing neurogenic muscle loss [8,12,27].

It is important to note that such molecular signatures can be present even in older individuals who engage in regular physical activity. This finding challenges the traditional perspective that distinguishes between “use” versus “disuse”, suggesting that sarcopenia is a consequence not solely from physical inactivity, but also from the intricate age-related changes in cellular signaling and tissue homeostasis [8] and may also be part of the lower-than-expected/hoped capability of physical exercise (even when performed systematically) to preserve muscle mass.

Although it remains debatable whether NMJ dysfunction precedes or follows sarcopenia, multiple studies consistently report an association between NMJ alterations and muscle atrophy. Normal and coordinated movement relies on the transmission of impulses from upper motor neurons in the motor cortex to lower motor neurons in the spinal cord, which then send signals along their axons to the NMJs. At this level, the release of acetylcholine into the synaptic cleft and its binding to postsynaptic receptors initiate the cascade that allows muscle contraction. With aging, motor neurons degenerate, NMJs become unstable—through the displacement of pre-terminal axons and a fragmentation of endplates—and mechanotransduction weakens, reducing contractile efficiency and limiting motor unit recruitment, in some cases, even before visible atrophy occurs. These alterations are further aggravated by inadequate protein intake or protein–energy malnutrition, which blunt anabolic responses despite apparently normal calorie intake [8,12,27].

#### 3.2.3. Muscle–Brain Crosstalk

Besides the structural atrophy, a complex endocrine dialogue between the skeletal muscle and the brain exists. Myokines, including irisin, BDNF, myostatin, pro-inflammatory cytokines, and chemokines, express neurotrophic and metabolic effects that protect the brain. However, aging influences this relationship, leading to reduced myokine production, weaker neuroprotection, and heightened inflammation and oxidative stress. In addition, decreased IGF-1 secretion can cause microvascular rarefaction, BBB disruption, and, in some cases, dementia. Thus, muscle atrophy can increase neurological vulnerability and vice versa, thereby establishing a cycle that connects sarcopenia, fragility, and neurodegenerative disorders [2,12,50].

#### 3.2.4. Muscle–Nerve Regeneration Failure and Extracellular Matrix Fibrosis

Beyond neural degeneration, the muscle microenvironment progressively loses its capacity to support reinnervation and repair. With aging, there is a decrease in the number and responsiveness of satellite cells, while skeletal muscle fibers exhibit dysregulated anabolic and catabolic signaling pathways. At the same time, the extracellular matrix (ECM), mainly produced by resident fibroblasts, undergoes maladaptive remodeling characterized by fibrosis, increased stiffness, and impaired mechanotransduction. These processes limit force transmission and physically obstruct reinnervation of denervated fibers. Consequently, several studies report a progressive reduction in muscle responsiveness to anabolic stimuli with aging [8,15].

Recent evidence has delineated a pivotal role for exerkines, defined as exercise-induced signaling molecules, that facilitate a multidirectional intercellular dialogue between muscle, nerve, and immune cells. According to a recent study, these molecules may provide promising therapeutic avenues for restoring neuromuscular homeostasis [8].

### 3.3. Cardiovascular Intermingles with Sarcopenia (Cardio-Sarcopenia)

#### 3.3.1. Arterial Hypertension (aHTN)

Progressive skeletal muscle wasting has been associated with reduced mobility and contributes to cardiovascular and metabolic disorders such as aHTN, insulin resistance, obesity and metabolic syndrome [2]. A study involving 2613 individuals found that aHTN was significantly associated with double the risk of sarcopenia in both genders (*p* < 0.001). Additionally, almost 60% of the patients used a minimum of 2 antihypertensive drugs. Moreover, patients with aHTN showed worse performance across nearly all sarcopenia-related functional tests compared to those without such disease [51].

#### 3.3.2. Heart Failure (HF)

Sarcopenia and HF represent two closely interlinked syndromes that share common molecular, metabolic, and inflammatory pathways and their coexistence is clinically important [2,45]. HF, defined as the heart’s inability to maintain an adequate cardiac output in order to meet metabolic demands, is overall the leading cause of morbidity and mortality after the age of 50, and its frequency is expected to rise with aging [8,11,45]. The management of HF often involves a multidisciplinary team that includes rehabilitation physicians and dietitians to develop an effective rehabilitation program and combat sarcopenia [52]. Structural and functional cardiac remodeling, along with systemic inflammaging and the accumulation of SASP (metabolically active cells that are unable to divide), progressively impairs cardiovascular resilience [8,45].

Interestingly, about one-third of HF patients meet the criteria for sarcopenia, with differences between sexes. For example, men tend to lose more muscle mass, while women often show a decline in functionality [3]. Those affected by sarcopenia typically have more extended hospital stays (studies underline that hospitalized individuals due to acute HF have the highest prevalence rates of sarcopenia, ranging from 34% to 66%), repeated admissions, and an increased risk of death [3,7]. Furthermore, approximately 47% of HF patients qualify as frail, regardless of age, highlighting a shared biological foundation rather than just the effects of chronological aging [16]. Both HF with reduced ejection fraction (HFrEF) and HF with preserved ejection fraction (HFpEF) are associated with muscle loss and sarcopenic obesity, which surprisingly occur frequently in HFpEF and contribute to exercise intolerance [11,16]. The Studies Investigating Comorbidities Aggravating Heart Failure (SICA)-HF study demonstrated that sarcopenia prevalence in HFrEF patients is nearly 20-fold higher than in age-matched controls [11], while the FRAGILE-HF registry reported sarcopenia in 23% of hospitalized HF patients ≥65 years old (interesting difference from the above-quoted prevalence of 34–66%), frequently coexisting with frailty, cachexia, or malnutrition, each contributing to increased mortality risk [3].

At the molecular level, the aged heart exhibits changes in metabolism, including increased fatty acid oxidation, insulin resistance, chronic inflammation, and oxidative stress, which contribute to reduced contractility and lipotoxicity. Dysfunction in cross-talk between the heart, skeletal muscle, liver, and adipose tissue has been associated with chronic pro-inflammatory conditions. Skeletal muscle can play a role as more than just a passive target; there are findings suggesting its involvement in heart failure-related pathophysiology by increasing the hemodynamic load on the heart. Catabolic myokines, such as myostatin, are associated with increased proteolytic stress, reduced satellite cell availability, and inhibition of the AKT/mTORC pathway, which can further affect heart function. In addition, IGF-1 deficiency has been associated with sarcopenia, premature atherosclerosis, endothelial dysfunction, increased inflammation, and plaque instability. Collectively, these observations indicate a close association between sarcopenia and HF and support the concept that muscle wasting is integrated within the broader biology of metabolic aging in HF populations, a pattern frequently observed alongside cardiac cachexia in advanced HF [7,9,45,50,52].

Frailty and polypharmacy magnify this interaction. In large geriatric cohorts, frail individuals with multiple chronic diseases showed a steep rise in adverse outcomes, with mortality risk multiplying when excessive polypharmacy (>10 drugs) coexisted. Falls and cardiovascular instability form another clinical intersection: orthostatic hypotension—often drug-related or linked to sarcopenia-associated autonomic dysfunction—is a preventable cause of recurrent falls and fragility fractures. Sarcopenia, arrhythmias, hypovolemia, anemia, and medications such as antihypertensives, antiarrhythmics, sedatives, and antianginals act synergistically to increase the risk of falls. This creates a vicious cycle among frailty, sarcopenia, and cardiovascular decline [16]. Furthermore, loop diuretics and reduced mobility contribute to calcium and vitamin D deficiency, which exacerbates bone loss, sarcopenia and increases fall risk [7].

#### 3.3.3. Sarcopenia and Other Cardiovascular Diseases

Sarcopenia and coronary artery disease (CAD) share core mechanisms of low-grade, chronic systemic inflammation. In community-dwelling older adults, low skeletal muscle mass correlates with subclinical atherosclerosis—higher coronary artery calcium burden, arterial stiffness, and carotid wall thickening—and sarcopenia may independently increase CAD risk [7]. A very recent study examining 318 older adults with established coronary heart disease reported that nearly one in five patients met diagnostic criteria for sarcopenia, and this subgroup displayed a distinctly more fragile cardiovascular profile. Individuals with sarcopenia showed lower cardiac output and reduced left-ventricular ejection fraction, together with a pattern of structural ventricular alterations that suggested impaired myocardial performance. These physiological disturbances were mirrored in clinical outcomes: adverse cardiac and cerebrovascular events, as well as short-term mortality, occurred more frequently among sarcopenic patients [53].

In patients with peripheral artery disease (PAD), and those with chronic limb ischemia in particular, sarcopenia occurs not only frequently, but also clinically relevant. Prospective cohort studies show that reduced skeletal muscle strength correlates with an increased incidence of serious adverse events and increased long-term mortality. These findings illustrate the importance of muscle strength in PAD [7].

#### 3.3.4. Sarcopenia and Cardiovascular Surgery

Sarcopenia is common among the elderly undergoing heart surgery and has been consistently associated with adverse postoperative outcomes and increased long-term mortality. Age, existing health problems, and poor nutrition—added to cardiovascular diseases—lead to muscle wasting and functional decline, creating a cycle of deterioration unless managed early [7].

Despite improved survival after Transcatheter Aortic Valve Replacement (TAVR) for severe aortic stenosis, sarcopenia remains common (~50%) and portends poorer functional recovery and health-related quality of life (HR-QoL) at 1-year post-TAVR, according to a study of 13,351 older adults. Sarcopenia, involuntary weight loss, and loss of the independence of performing ADL were, according to a study, significant factors that lead to suboptimal outcomes. These data highlight the need to systematically address sarcopenia within TAVR treatment approaches [7].

Heart transplantation remains the standard of care for end-stage HF, yet persistent exercise intolerance and skeletal myopathy are common due to chronic and necessary immunosuppression, cardiac denervation, metabolic complications, and antecedent HF-related muscle wasting. Post-transplant sarcopenia is associated with an increased number of complications, hospitalization length of stay, and mortality. Recent European guidelines (European Association of Preventive Cardiology; Heart Failure Association of the European Society of Cardiology; European Cardio-Thoracic Transplant Association) emphasize that current rehabilitation frameworks inadequately target post-heart transplantation sarcopenia; yet, such patients need rehabilitation because they frequently have myopathy, reduced aerobic efficiency, fiber atrophy, and impaired capillarization, often exacerbated by corticosteroids, malnutrition, and inactivity. Therefore, proactive, structured, multidisciplinary rehabilitation that specifically screens and treats sarcopenia is crucial for improving functional independence and survival [7,54].

### 3.4. Sarcopenia in Children—Is There a Connection?

Although sarcopenia is traditionally regarded as an age-related condition, emerging evidence suggests that key biological mechanisms involved in muscle loss and dysfunction may also be relevant earlier in life. In this context, pediatric sarcopenia is discussed here not as a distinct clinical entity, but as a conceptual extension of the sarcopenia spectrum, highlighting biological continuity across the lifespan.

Sarcopenia in children is considered to have a negative impact on growth and neurocognitive development, but, at the present time, there are limited studies that have evaluated the impact of child sarcopenia. In children, sarcopenia was only recently described in relation to malnutrition and reduced skeletal muscle mass and there is still a lack of uniform definition and diagnostic criteria. A review published in 2020 concluded that sarcopenia enhanced the risk of fungal infection in children with acute conditions and also the overall inpatient length of stay. The same article specifies that there is no current golden standard method to assess motor function impairment in children with sarcopenia, which makes the definition and classification of the disease in children more difficult. It is clear that poor muscle function affects fine and gross motor, as well as cognitive, development in early childhood. It is considered that malnutrition plays a key role in sarcopenia development in children, along with physical inactivity and hypermetabolism. For example, vitamin D and protein deficiencies, combined with reduced weight-bearing activities during childhood, can contribute to the development of sarcopenia in adulthood [55].

A review published in 2024, which included 56 studies, tried to define pediatric sarcopenia and its implications. The review concluded that most of the studies they cited used CT or MRI to assess skeletal muscle mass indicators, such as total psoas muscle area, along with the associated risks of sarcopenia for various clinical outcomes. The same research stated that, in children, bone mineral density is correlated with skeletal muscle mass and associated with cardiovascular and metabolic diseases. Moreover, lower grip strength was observed in adolescents with metabolic syndrome, suggesting that skeletal muscle mass loss and poor muscle strength in the pediatric period may mark other health problems in adult life. It is believed that several factors contribute to the loss of muscle mass and strength, including vitamin D deficiency, disease-related inflammatory cytokines, steroid treatments, and physical inactivity. Rather peculiar, reduced muscle strength in adolescence was identified as a predictor of disability three decades later in one study included in the aforementioned review, and, in another, as a risk factor for all-cause mortality within 24 years, being associated with cardiometabolic disease, bone and neurodevelopmental deficits. Despite the implications, reduced skeletal muscle mass, poor muscle strength, and sarcopenia in pediatrics lack well-established definitions. Therefore, clear definitions, classification and diagnostic criteria for these health problems in children have to be assessed in the near future [56].

A systematic review from 2025 evaluated sarcopenia in children with various types of cancer by reviewing articles published over the last 5 years. It appears that sarcopenia in children with cancer negatively impacts their recovery, predisposes them to different infections and can also affect the success of cancer treatment. Sarcopenia can be present both during and after treatment, before surgery or chemotherapy and can carry on long after the end of the treatment. The study showed that several factors appear to be involved in the development of sarcopenia in pediatric tumors. For example, a lack of physical activity can impair mitochondrial muscle metabolism, as well as chemotherapy, which reduces appetite and caloric intake, thereby enhancing malnutrition. Moreover, growth seems to be disturbed by mutations in the PI3K/AKT/mTOR signaling pathway and cisplatin treatment stimulates the expression of NF-κB in muscles, causing their mass loss. In addition, sarcopenia that appeared after cancer treatment, especially within the first year of diagnosis, seemed to affect antitumor immunity by blocking the immune cells from entering the tumor, aggravating the outcomes. The study concluded that sarcopenia is a clinically important condition in pediatric cancer patients and the prevention of sarcopenia can improve the survival rate [57].

Viewed through this lens, pediatric sarcopenia reinforces the concept of sarcopenia as a multisystem condition involving muscle–neural–metabolic interactions, rather than a process confined to chronological aging.

### 3.5. Diagnosing Sarcopenia

The diagnosis of this condition involves multiple layers and is based on the integrated evaluation of muscle mass, strength, and physical performance, reflecting the contemporary shift from anatomical assessment toward functional and clinical endpoints. In practice, case detection is often triggered by clinical suspicion—particularly in individuals presenting with unintentional weight loss, generalized weakness, impaired balance, or recurrent falls—even in the absence of overt muscle atrophy. Recognition of risk-enhancing factors, such as prolonged glucocorticoid exposure, endocrine disturbances, chronic inflammatory diseases, malignancy, cognitive impairment, polypharmacy (including immunosuppressants, if applicable), dysphagia, or recent hospitalization, is essential for identifying at-risk patients [2].

Screening typically begins with simple, pragmatic bedside tools. A self-administered questionnaire, assessing self-reported Strength, Assistance with walking, Rise from a chair, Climb stairs and Falls (acronym SARC-F), although highly specific, shows limited sensitivity and may not detect early or mild cases. Anthropometric and performance-based screening methods recommended by recent guidelines—such as measuring calf circumference or applying the finger-ring test, chair stand test, Timed Up-and-Go, handgrip strength estimation, and gait speed—are widely used in both community and clinical settings. The concept of “functional sarcopenia,” i.e., low strength and performance with preserved muscle mass, highlights the importance of clinical functionality over morphology alone in guiding early diagnosis [2,11,47].

Muscle strength is most commonly quantified using handgrip dynamometry, with values below 27 kg in men and 16 kg in women considered diagnostic for clinically relevant weakness. Unfortunately, this clinical method does not assess overall sarcopenia. Physical performance is evaluated using gait speed over 4–6 m (<1.0 m/s, the endorsed threshold) or the five-time chair stand test, in which impaired function is defined as completion times of 10–12 s or more. Composite instruments such as the SPPB, which incorporates balance, gait speed, and chair rise performance into a 12-point scale, are widely used to stage severity. Scores below 9 indicate reduced functional status [2,47].

Assessment of muscle mass most frequently relies on DXA or BIA, with appendicular skeletal muscle mass (ASMM) indexed to height squared (kg/m^2^), as recommended by the AWGS and Korean Working Group on Sarcopenia (KWGS). Nevertheless, both DXA and BIA remain indirect measures that have a variable association with functional outcomes. Imaging techniques, such as CT, MRI and US, may provide a detailed assessment of muscle quality and fat infiltration, but are not used routinely because of the high cost [2,47].

The D3-creatine dilution method offers the advantage of directly and more precisely quantifying muscle mass and can therefore improve diagnostic accuracy. The method is based on oral intake of deuterated creatine and subsequent 24 h urine collection to quantify D_3_-creatinine excretion [2,47,58].

Beyond imaging and function, circulating biomarkers are increasingly explored. C-reactive protein (CRP), a widely available biomarker of systemic inflammation, has also gained renewed interest, as elevated CRP levels correlate with inflammaging and greater sarcopenia severity—more specifically with decreased handgrip and knee extension strength; however, recent analyses emphasize that CRP should be used only as part of an integrated diagnostic algorithm rather than as an isolated marker, due to its lack of disease specificity and susceptibility to false-positive results in multimorbid patients [59].

The serum cystatin C-to-creatinine (SCr) ratio has shown promise; elevated values of this biomarker reflect a disproportionate loss of muscle mass relative to renal filtration. The sarcopenia index (SI), which combines cystatin C with the estimated glomerular filtration rate, has also been studied. Although controversial, other potential diagnostic candidates include trace elements like manganese and markers of intestinal permeability such as diamine oxidase [26].

### 3.6. How Can We Interfere with the Progression of Sarcopenia?

The evidence from intervention trials identified through the literature search was considerably more limited and heterogeneous than that from mechanistic and diagnostic trials. Therefore, this next step of the review is also presented primarily through narrative means and an integrative approach, centered on mechanistic reasoning and consistency among intervention trials, rather than further pursuing quantitative synthesis.

Sarcopenia diagnosis must be followed by prompt intervention to prevent progressive morpho-functional decline. Current management strategies focus on both pharmacological and non-pharmacological approaches. Although sarcopenia mainly affects older adults, the mechanisms behind muscle and bone loss are not exclusive to aging. An interesting example comes from space medicine: young and physically fit astronauts experience a rapid loss of muscle and cardiovascular fitness during long periods in microgravity. This environment is associated with accelerated muscle wasting, decreased bone density, and postural stability difficulties, changes that typically develop more gradually in sarcopenia. Research on interventional strategies to counteract sarcopenia and its consequences includes high-intensity interval training (HIIT), mechanical loading devices, vibration therapy, neuromuscular electrical stimulation, and nutritional optimization [12,19,20,23,44].

#### 3.6.1. Pharmacokinetics and Pharmacological Modulation of Sarcopenia

##### Established and Repurposed Therapeutic Strategies

Despite years of research, no single drug has been confirmed as a standalone treatment for sarcopenia. This highlights the need for a combined and integrative approach [2]. Correction of vitamin D deficiency is widely regarded as clinically relevant, as low serum levels are associated with upregulation of myostatin, diminished muscle mass, reduced strength, increased fall risk, fragility fractures, cognitive impairment and, respectively, poorer hospitalization outcomes [2,12,15,17,43,60]. Regular supplementation of 800–1000 IU daily seems helpful for older adults with VD deficit and also diminishes the risk of falls by 19%. In contrast, high doses are not recommended for those without deficiencies. The exact role of VD in sarcopenia remains unclear due to a lack of focused clinical trials [21,22,43].

It may be noteworthy to emphasize that commonly used cardiometabolic drugs often have positive effects on muscle metabolism. Angiotensin-converting enzyme inhibitors (ACE-Is), angiotensin receptor blockers (ARBs), and β-blockers can improve blood flow, mitochondrial function, and reduce inflammation. Beyond managing blood pressure, ACE-Is have been reported to influence IGF-1 signaling and muscle-related outcomes in some clinical cohorts. They also promote angiogenesis and improve endothelial function, possibly delaying the onset of sarcopenia. However, frail or sarcopenic hypertensive patients frequently receive suboptimal therapy despite the potential for greater benefits [3]. Additionally, the SARcopenia Assessment in Hypertension study, which included 1775 patients with aHTN, demonstrated that among female participants, the subset using ACE-Is exhibited higher grip strength and better chair-stand test performance, whereas those treated with ARBs showed greater anterior thigh muscle thickness and gait speed [51].

Other drugs that affect metabolism, such as sodium-glucose cotransporter-2 (SGLT2) inhibitors, statins, glucagon-like peptide-1 receptor agonists, 3-hydroxy-3-methylglutaryl coenzyme A, and dipeptidyl peptidase 4 inhibitors, have been investigated for additional anti-inflammatory and metabolic benefits, though current data are heterogeneous [22].

Metformin, although primarily an antidiabetic medication, has attracted attention as a geroprotective and exercise-sensitizing agent. Experimental data suggest that, under certain biological conditions—such as non-diabetic metabolic status, age-related endocrine imbalance, altered redox states, dysregulated mitochondrial or AMPK–mTOR signaling pathways, or in the absence of concurrent physical exercise—metformin might negatively affect muscle function by increasing myostatin and altering anabolic signaling. However, some studies show that when metformin is used alongside exercise or focuses on oxidative stress and metabolic balance, it can lead to muscle mass growth, increased strength, and functional abilities, as well as protection against muscle loss with aging. Human research is limited and heterogeneous, necessitating further study before conclusions about metformin’s effects on sarcopenia can be drawn [22].

Hormonal treatments play a supportive but cautious role. Growth hormone (GH) and its mediator, IGF-1, aid in protein production and satellite cell activation. GH deficiency correlates with higher myostatin levels and impaired muscle recovery, but its supplementation, however, does not fully restore normal protein synthesis. Tesamorelin, a GH-releasing hormone analogue, has been found to enhance muscle density and lower body fat in patients with human immunodeficiency virus-associated lipodystrophy [17,22,50]. Some animal studies show that IGF-1 treatment also improves cerebrovascular angiogenesis [50]. Additionally, supplementation with testosterone or estrogen can increase muscle mass and strength, but may also heighten cardiovascular and oncologic risk, needing careful individual assessment [17].

A study reports that exercise, paired with endogenous IGF-1 stimulation (although its activation is generally controversial), is linked to improvements in microvascular structure and perfusion in skeletal muscle and brain tissue, showing bidirectional benefits [50].

Nutraceuticals and dietary supplements serve as practical adjuncts to pharmacological therapy. Protein, essential amino acids, creatine, calcium, and antioxidants (D, C, E) enhance anabolism and recovery. Polyphenols, such as resveratrol (that also activates SIRT), quercetin, and epigallocatechin-3-gallate (EGCG) stimulate PGC–1α–mediated mitochondrial biogenesis and reduce oxidative stress. Coenzyme Q10 and L-carnitine—already used in clinical practice—support mitochondrial metabolism and muscle endurance, while ginsenosides and quercetin have been linked to improved skeletal muscle contractility [6,12,22,42].

##### Experimental and Emerging Pharmacological Strategies

Starting about two decades ago, and hopefully more successfully in the quite recent years, some monoclonal antibody-type pharmacological agents have been investigated for their influence on muscle metabolism and protein turnover. In this respect, myostatin inhibitors such as Bimagrumab, Stamulumab, Trevogrumab, and Landogrozumab have demonstrated increases in lean mass, although functional gains remain inconsistent. Selective androgen receptor modulators (SARMs), such as enobosarm, have shown greater promise, with early trials reporting improvements in both lean body mass and functional performance. Exercise mimetics, including PPAR-δ agonists and AMPK activators such as AICAR and metformin, aim to chemically mimic the benefits of exercise, although they have not yet reached the market [2,9].

A Phase I clinical trial of BIO101 (20-hydroxyecdysone), a naturally derived steroid, showed good safety and potential for improving mitochondrial function, making it an encouraging treatment option for sarcopenia that requires larger trials to validate [20].

Compounds such as resveratrol, Mitoquinone Q, and Szeto–Schiller peptides (SS-31) aid the growth of mitochondria and their balanced energy use. Likewise, AICAR, sulforaphane, IL-1 inhibitors, and SRT2104 have demonstrated protective effects against age-related muscle loss [9,14]. The Targeting Aging with Metformin (TAME) group emphasizes the lack of validated human biomarkers that accurately reflect biological aging; in this respect, the authors proposed several biomarker options for monitoring the efficacy of treatments, such as IL-6, TNF-α receptors I/II, CRP, Growth Differentiation Factor (GDF)15, insulin, IGF-1, cystatin C, N-terminal pro-B-type natriuretic peptide, and glycated hemoglobin. Further research on mitochondrial proteins may help close this gap [7].

Gene therapies focused on mitochondrial regulation and function, such as increasing the levels of genes like PGC-1α, SIRT1, and Parkin, may prevent muscle wasting by improving mitochondrial health. Non-coding RNAs, like lncRNA EDCH1 and lncRNA Pvt1, also present potential as treatment targets for muscle loss. Other approaches, including extracellular vesicle therapy from stem cells, heat stress, electrical stimulation, and caloric restriction, also boost mitochondrial health and prevent muscle atrophy. These interventions can be paired for the best treatment outcomes. Moreover, natural substances, for example: urolithin A, Lactobacillus paracasei PS23, Gomisin G, and Ginsenoside Rg3 support muscle regeneration and mitophagy in various preclinical models. Nanomedicine strategies like Se@SiO2 nanoparticles and CoQ10 nanodisks aim to modulate mitochondrial stability and reduce oxidative stress. However, their clinical use faces challenges with safety and scalability [14].

Emerging translational strategies—including senotherapeutics, microbiota modulation with probiotics and prebiotics, and soluble klotho, a circulating anti-aging hormone shown to counteract TGF-β–induced myogenesis impairment—further reflect a shift toward disease-modifying interventions that aim to restore regenerative capacity rather than solely counteract atrophy [15,48].

#### 3.6.2. Non-Pharmacological Interventions for Sarcopenia

##### Nutrition and Gut Microbiota Modulation

Nutritional optimization represents the cornerstone of sarcopenia prevention and management, especially when it complements sustained physical activity. This strategy has been associated with improvements in microbial diversity and muscle-related outcomes in several studies [2,26]. The International Clinical Practice Guidelines for Sarcopenia (ICFSR) recommend that older adults should consume between 1.2 and 1.5 g of protein per kilogram of body weight each day. The recommendations also stress the importance of good-quality proteins, particularly those rich in leucine, which are known to stimulate mTORC1 and IGF-1 signaling pathways and thereby support muscle protein synthesis [2]. Interestingly, Heyland et al. [3] conducted a randomized, multicenter study on critically ill patients that underlined the benefit of diet: greater habitual protein intake, particularly from animal sources, was associated with superior lean mass conservation in the elderly, and individualized nutritional support during hospitalization decreased mortality and cardiovascular events in malnourished HF patients.

Harmonized consensus statements published by the European Society for Clinical Nutrition and Metabolism (ESPEN) further refine these recommendations: for healthy older adults, the habitual protein intake should be 1.0–1.2 g/kg/day and further increased to 1.2–1.5 g/kg/day and even higher in states of severe catabolism. It is important to note that these guidelines also outline the synergistic combination of resistance and aerobic exercises, which results in maximum anabolic response. Protein source and digestion kinetics also play a role: whey and soy (“fast” proteins) result in prompter amino acid release and greater synthesis after RT than does casein (“slow” protein), as Tang et al. have emphasized [7].

Micronutrient and antioxidant intake complements macronutrient optimization, although the evidence is still inconclusive. Antioxidants such as EGCG, curcumin, and resveratrol improve muscle metabolism and physical performance by modulating BDNF expression (which acts as an important muscle regenerator and directly impacts the NMJ) and reducing oxidative stress [12,17]. Diets rich in good-quality proteins (such as dairy, fish, meat, and soy), omega-3 fatty acids, fibers, and polyphenols, in conjunction with adequate intake of vitamins D, C, and E, are associated with the preservation of skeletal muscle mass, bone mineralization, and cardiac and cognitive function [5,26,45]. Nutritional counseling by trained dietitians has been shown to improve QoL and may even reduce mortality in populations with sarcopenia and HF [41,52].

Emerging research has identified the gut–muscle axis as a key modulator of systemic and skeletal muscle homeostasis. In chronic kidney disease—a prototypical high-risk state for sarcopenia—intestinal dysbiosis, characterized by reduced Lactobacillus and Bifidobacterium species and overgrowth of uremic toxin–producing bacteria, contributes to mitochondrial oxidative stress and muscle catabolism. Supplementation with pre-, pro-, and synbiotics is associated with improved muscle performance and a lower risk of sarcopenia. Restoring butyrate-producing bacterial populations is emerging as a mechanistic target in microbiome-directed therapy for sarcopenia, but interestingly, also for myocardial fibrosis [15,26,61].

##### Physical and Exercise-Based Interventions

Exercise is increasingly investigated as a biologically targeted intervention with potential disease-modifying effects capable of reversing the core molecular mechanisms of sarcopenia, yet outcomes vary widely depending on training modality, intensity, and physiological context—indicating that not all exercise stimuli elicit equivalent biological responses. This has led to a growing recognition that exercise must be prescribed with mechanistic precision, rather than general lifestyle advice. Habitual physical activity reduces the risk of falling, counteracts mitochondrial dysfunction, increases BDNF levels, improves cognitive function, hippocampal activity, cardiac remodeling, depressive symptoms (also a risk factor for sarcopenia, falls, and fragility fractures through the cortisol axis), and QoL [18,43,45,62].

The included intervention studies in this systematic review differed substantially in terms of exercise modality (resistance, aerobic, combined), training dose, supervision level, and outcome assessment tools, limiting direct comparison across studies.

Mitochondrial and Molecular Mechanisms, Targeted

Exercise enhances mitochondrial quality and turnover through PGC-1α-, PI3K-, and Nuclear Factor Erythroid-Related Factor (NRF)1-, and NRF2-driven biogenesis and mitophagy activation, counteracting the mitochondrial dysfunction central to aging muscle [12,14]. Aerobic training leads to oxidative phosphorylation, while RT boosts anabolic signaling and mitochondrial resilience. However, excessive or poorly managed exercise can temporarily increase oxidative stress. Therefore, a moderate and personalized intensity is crucial, especially for frail or multimorbid patients [42,45]. Complementary interventions, including nutritional optimization, electrical stimulation, heat therapy, and caloric restriction, further enhance mitochondrial health [14].

These and other strategies are being applied in prehabilitation—both before and after major physiological stressors—with demonstrated enhancement in resilience and recovery. Personalized exercise programs, calibrated to biological age, mitochondrial capacity, and mitophagy status, represent perhaps the most promising future direction in sarcopenia care [42].

A recent systematic review and meta-analysis evaluated 12 randomized controlled trials involving 827 previously inactive adults and concluded that exercise interventions lasting from 6 weeks up to 12 months can significantly modulate DNA methylation patterns, both globally and at gene-specific sites. While changes varied by study, key findings included reduced global methylation in some cohorts and increased methylation of specific genes in others. Knowing that aging muscle is characterized by a progressive change in DNA methylation patterns, these findings are clinically important, highlighting the fact that epigenetic adaptation may contribute to the beneficial health effects of exercise and reinforce the importance of optimizing exercise dose, modality and duration when addressing age-related or metabolic disorders in which epigenetic dysregulation plays a role [63]. Additionally, recent Romanian data highlight a genetic dimension in sarcopenia susceptibility. Nedelcu et al. reported that homozygosity for the GDF5 rs143384 risk allele (AA) was more prevalent among sarcopenic and sarcopenic obese patients, whereas the G allele appeared protective. Importantly, a twelve-day rehabilitation program led to significant reductions in SARC-F scores across both genetic risk groups, suggesting that functional improvement is achievable irrespective of polymorphism status. These findings complement evidence on exercise-induced epigenetic adaptations and further support the rationale for personalized rehabilitation strategies that consider molecular susceptibility [64].

Beyond hypertrophy, physical activity enhances NMJ stability, improves motor unit recruitment, restores mitochondrial biogenesis, and mitigates chronic inflammation and AGE accumulation. Even in the absence of hypertrophy, adaptive remodeling of NMJs and motor units enhances muscle performance—a crucial compensatory mechanism in advanced sarcopenia or during immobilization, where strength may decline by up to 1% per day of bed rest [2,6,44]. Exercise also improves microvascular perfusion and stimulates the secretion of exerkines, thereby restoring inter-organ communication between skeletal muscle, brain, adipose tissue, and the immune system. This places exercise among the few interventions with simultaneous musculoskeletal, cardiometabolic, neuroprotective, and anti-inflammatory effects, targeting multiple biological pathways implicated in the multisystem nature of sarcopenia [25,65].

2.Resistance and Multimodal Training: The Therapeutic Core

Among all modalities, progressive RT remains the gold standard for sarcopenia treatment and prevention. RT selectively targets type II (fast-twitch) fibers—as aforementioned, those most affected by aging—improving motor unit synchronization, firing frequency, and joint stability [10,15,19,47,61,66]. Standard protocols of 1–3 sets of 6–15 repetitions at 30–70% of the intensity of one-repetition maximum, performed 1 to even 6 times weekly—although most studies were performed with an average of 3 RT sessions/week—can result in improvements in muscle mass and strength even in individuals over 75 years old [66].

In a meta-analysis including 2,485 middle-aged and older adults with sarcopenia, RT alone produced a mean difference (MD) in handgrip strength of 2.58 kg compared with control. Moreover, ASMM increased by 0.90 kg, and walking speed improved by 0.28 m/s. The combined intervention of RT and nutritional support yielded the greatest gains: fat-free mass improved by 5.15 kg, Timed Up and Go (TUG) time reduced by 2.31 s and Chair Stand Test time by 2.37 s. These findings underscore the role of RT in reversing functional decline and maintaining independence [19].

Aerobic (endurance) training provides essential complementary effects, improving oxidative metabolism, endothelial function, and balance-based autonomy. It may also help preserve NMJ integrity while reducing low-grade inflammation and metabolic stress. Aerobic training alone, however, cannot prevent sarcopenia in the elderly; thus, combined programs integrating resistance, aerobic, balance, and motor–cognitive components provide the most comprehensive benefits, reinforced by appropriate nutrition as well. Moreover, these interventions can favorably modulate microbiota [8,21,61].

Motor–cognitive training—combining simultaneous physical and cognitive challenges—has shown superior efficacy in preventing sarcopenia, as well as frailty, falls, and cognitive decline, reinforcing the understanding of sarcopenia, including from integrated functionality (propensive to autonomy) as both a neuromuscular and neurocognitive disorder shaped by muscle–brain communication [27,66].

Regarding cardiovascular pathology, sustained physical activity exerts a protective effect against ischemia–reperfusion injuries, prevents arterial atherogenesis, regulates the autonomous activity of the body structures and functions, and stimulates myocardial regeneration [6,17]. Cardiac rehabilitation is able to address risk factors, accelerating the development of muscle-wasting disorders, and also improve muscle mass and strength [7].

Long-term rehabilitation programs lasting more than 12 months have additional benefits for survival and hospitalization rates, especially in patients with HFrEF [11]. In patients who receive a heart transplant, rehabilitation programs combining resistance, endurance, and inspiratory muscle training have helped such individuals to improve peak oxygen volume, insulin sensitivity, bone mineral density, and microvascular function, thus preventing further muscle wasting caused by immunosuppression and sarcopenia [54].

Furer and Hawley propose the concept of “molecular athlete”, in which neuromuscular adaptation is a result of the deliberate activation of key metabolic and anabolic pathways, including mTORC1 for hypertrophy, AMPK–PGC-1α for mitochondrial resilience, and exerkine signaling for regeneration. The theoretical framework identified supports the idea of molecular periodization, consisting of timing exercise and nutrition, and manipulating exercise intensity to surmount anabolic resistance and maximize therapeutic effect. Moreover, older, but physically active individuals with an extensive athletic background have fewer denervated muscle fibers and a better ability for their reinnervation, which implies that exercise may help reduce the rate at which sarcopenia progresses [62].

Monti and colleagues assessed the long-term impact of a 2-year multimodal exercise intervention comprising progressive RT and functional balance and coordination exercises on aging muscle in older adults with poor muscle function. The participants achieved remarkable improvements in muscle strength, as objectively assessed with specific tests and scales (SPPB, chair-stand) and preserved muscle architecture of the vastus lateralis over the 2 years, preserved NMJ stability, as reflected by the constant CAF levels—a marker of NMJ degradation [27].

3.Exerkines and Systemic Crosstalk Contribution

Sustained physical activity mimics hormone-like effects through the secretion of exerkines, which act via auto-, para-, and endocrine pathways to regulate metabolic, anti-inflammatory, and neuroprotective responses. Kwon et al. demonstrated that individuals with Parkinson’s disease have a higher prevalence of Parkinson’s disease and altered expression of key myokines, including apelin, beta-aminoisobutyric acid (BAIBA), IGF-1, irisin, sestrin, IL-15, and the vascular endothelial growth factor (VEGF), alongside increased pro-catabolic mediators such as IL-6 and myostatin. Aerobic exercise preferentially upregulated apelin, BAIBA, and VEGF. Additionally, RT enhanced Bone morphogenetic protein 7 (BMP-7), IGF-1, and decorin. All these molecules can act beneficially by suppressing myostatin expression [65].

This evidence underlines the systemic effects of physical training in muscular, metabolic, and neurocognitive areas, a hypothesis that also has received support from studies indicating that higher levels of physical activity are associated with lower circulating levels of the neurodegeneration biomarker NfL [27].

4.Digital Rehabilitation and Post-Pandemic Opportunities

The COVID-19 outbreak led to widespread isolation among frail individuals, accelerating functional decline, sarcopenia, depression, metabolic dysregulation, and fall risk. In this context, virtual and home-based technologies have become important and useful extensions of rehabilitation [12,18].

Telehealth, as defined by the WHO, is the remote delivery of healthcare via information and communication technologies, and it has rapidly expanded in recent years as a means to overcome limited access to in-person care. There is a strong opportunity to improve the delivery of exercise and rehabilitation programs for older adults in aged care facilities, where mobility decline and falls are pervasive concerns. Yet, evidence with regard to the effectiveness and optimal implementation of such programs specifically for frail high-dependency aged care populations remains limited, and seemingly the same goes for sarcopenia. However, a recent systematic review and meta-analysis conducted according to PRISMA guidelines evaluated exercise-based telehealth interventions in older adults (“60+ years”), including those with frailty, mobility, cognitive impairments, or receiving aged care services at home or in residential facilities. Hospital-based populations were excluded from this review to ensure the findings are relevant to community and aged-care settings. The authors also reviewed the effects on mobility, balance, muscle strength, incidence of falls, and the general QoL, using a variety of validated measures, including the SPPB, TUG test, walking speed assessments, the Berg Balance Scale, and the timed Sit-to-Stand test. The authors reported that telehealth-delivered exercise interventions are feasible, effective, and safe for older adults with frailty, mobility, or cognitive impairments, with high adherence and minimal adverse events. Although the overall certainty of evidence was low, the results indicated small to moderate improvements in mobility and lower-limb strength, and a modest positive effect on balance, with no significant change in the QoL [1].

Virtual reality (VR) and exergaming platforms—such as Wii Fit and Kinect—combine play with structured movement, improving balance, reaction time, and coordination while maintaining adherence and motivation. This refers, including/especially to the elderly who, as is well-known and aforementioned, are prone to sarcopenia, too. Overall, the findings suggest that technology-supported exercise carried out at home can be both feasible and beneficial for maintaining independence and well-being in older adults. A randomized controlled trial conducted by Lee (2023) [18] examined the effects of a home-based exergame program (a combination of exercise and video games) in 57 community-dwelling adults aged 75 years or older from Seoul. All participants received health education sessions on musculoskeletal health and the experimental group also performed Ring Fit Adventure exercises on the Nintendo Switch three times a week for eight weeks, at home. The researchers assessed several outcomes, including physical function, confidence in avoiding falls (modified Falls Efficacy Scale), depression, and health-related QoL. In this respect, participants who underwent the exergame program showed statistically significant progress in balance, mobility, and lower-limb strength, along with a reduction in depressive symptoms and improvement in perceptions of physical health.

Recent studies have highlighted the strong influence of social connectivity on the adoption of digital therapeutics among older adults. Gamified interventions, examples of which include Sarcopenia Integrated Management and Measurement Systems (SIMMS), that unite medical wearable sensors with interactive exercise platforms, could simultaneously address physical training, motivation, and social engagement. Such strategies can be seen as hybrid medical and behavioral interventions that provide both rehabilitation and active aging [28].

Additionally, meta-analyses confirm that VR-based and telehealth exercise programs significantly enhance mobility, strength, and adherence, with high safety and feasibility in frail populations and a particular benefit for sarcopenic HF patients [1,3,18,28].

## 4. Discussion

To our knowledge, this is the first PRISMA systematic review that comprehensively integrates the neurological and cardiovascular perspectives on sarcopenia, while also detailing shared pathophysiological mechanisms, diagnostic strategies, and treatment, including a consistent rehabilitation approach.

Sarcopenia is a complex, heterogeneous condition shaped by the convergent influences of aging, general biologic involution, chronic low-grade inflammation, impaired NMJ morpho-functional connectivity, and decreased mechanical loading. Evidence from clinical and experimental studies indicates that its pathophysiology extends well beyond a simple reduction in muscle mass or strength. Mitochondrial dysfunction, anabolic resistance, oxidative stress, and persistent low-grade inflammation interact to compromise neuromuscular integrity and metabolic flexibility [2,7,14,22,40]. These processes offer a plausible biological framework for the frequent coexistence of sarcopenia with frailty, cognitive decline, and cardiovascular dysfunction, which together form an interdependent, multisystem syndrome rather than separate disease entities [2,5,16].

So, accumulating evidence suggests that sarcopenia should not be regarded solely as an isolated muscular disorder with clinical, functional, and on-QoL convergent impact, as it evolves, as mentioned above, in close association with metabolic, cardiovascular, and neurological decline.

Importantly, although rehabilitation and exercise-based interventions are widely regarded as central components of sarcopenia management; however, the strength of evidence supporting specific protocols remains uneven. Most available interventional studies are heterogeneous in design, sample size, training modality, dose, supervision, and outcome selection, which limits direct comparability and precludes definitive conclusions regarding the superiority of one approach over another. For this reason, rehabilitation-related findings are interpreted in a critical and integrative manner, emphasizing biological plausibility, consistency of directionality across studies, and translational relevance rather than hierarchical effectiveness rankings.

Beyond the international data synthesized above, the Romanian context also illustrates both the potential and the challenges of telemedicine in older populations. In a recent pilot survey, Aurelian and colleagues explored physicians’, trainees’, and elderly patients’ perceptions of e-health tools for cardiovascular risk assessment and prevention, showing that although digital health and telemedicine are theoretically well accepted, nearly half of older respondents reported low confidence in using teleconsultation platforms. This pattern mirrors broader concerns about digital health equity and underlines the need for structured training of both patients and providers before tele-rehabilitation can be scaled in geriatric practice [67]. At the same time, randomized trials in older adults with sarcopenia have demonstrated that app-based or artificial intelligence-supported RT programs can yield improvements in muscle strength and balance, suggesting that well-designed telerehabilitation may be a viable alternative for patients with limited access to conventional services. Likewise, recent systematic reviews indicate that home-based tele-exercise can improve muscle mass, strength, and physical performance in older adults with, or at risk of, sarcopenia, although the certainty of the evidence remains moderate and protocol heterogeneity is substantial. Taken together with our findings, these data support the development of hybrid, telehealth-delivered multimodal exercise programs—possibly combined with wearable sensors and gamified platforms—as a pragmatic way to maintain continuity of care for sarcopenic, frail, or long-COVID patients, provided that infrastructural barriers and the marked digital literacy gap among older people, as documented in Romanian surveys, are explicitly addressed [68,69,70,71,72].

From a clinical perspective, the findings summarized in this review support a multidimensional approach to sarcopenia assessment and management. Rather than being driven by isolated muscle loss, sarcopenia reflects the interaction of neuromuscular, neural, cardiovascular, metabolic, and inflammatory processes, which may manifest variably across patients and clinical settings.

In practice, this underscores the importance of combining functional evaluation with body composition assessment and contextual clinical information. While no single diagnostic tool captures the full biological complexity of sarcopenia, integrating simple performance measures with imaging, laboratory data, and comorbidity profiles may improve risk stratification and guide individualized rehabilitation strategies. Importantly, many proposed interventions remain supportive rather than disease-modifying and should be interpreted in light of the current evidence base.

To facilitate clinical interpretation, Table 1 summarizes the main mechanistic domains discussed in this review and links them to their potential clinical relevance, commonly used assessment tools, and broad intervention domains.

Despite such advances, tough challenges remain. Diagnostic criteria and threshold definitions are still heterogeneous, meaning comparability between studies is limited, and early detection can be hampered [2,3,8,19,20,23]. While candidate biomarkers like myostatin, IGF-1, cystatin C, and CRP show promise, these ones have not yet achieved uniformity in clinical use [7,26,59].

The present literature review has several inherent limitations that should be acknowledged. First, although the search was conducted systematically, restricting inclusion to English-language, full-text, ISI-indexed publications may still have excluded relevant data from non-indexed or non-English sources. This limitation was counteracted by adding some freely found sources with the aim of enhancing the quality of the manuscript. Considerable heterogeneity exists across the included studies—in diagnostic criteria, imaging or functional assessment tools, and exercise or rehabilitation protocols—which limits direct comparability and precludes a quantitative synthesis from their contents. In addition, the certainty of evidence is constrained by the predominance of observational and cross-sectional study designs, small or selected study populations, and frequent reliance on surrogate or indirect outcome measures. Although a formal assessment of publication bias (e.g., funnel plot analysis) was considered, this was not feasible due to the absence of multiple independent, comparable interventional studies reporting harmonized quantitative effect estimates with measures of precision. These limitations should be considered when interpreting the strength and generalizability of the present findings. Finally, despite a focus on and promises foreseen for the (newer) digital interventions, overall evidence in this area remains scant and uneven, with a need for larger, rigorously designed trials.

Future directions should focus on personalized approaches, including consideration of each individual’s physical abilities, as well as precision-based initiatives that incorporate biomarker profiling, wearable technology, and personalized (including digital) rehabilitation, and frame them within inclusivity within the P4 Medicine concept [10,73,74,75]. Whether implemented in supervised clinical environments or through telehealth and exergaming platforms, the core philosophy remains consistent: exercise is medicine and, more importantly, is essential for physical and mental health and social engagement, hallmarks of healthy and active aging [3,18,25,62,65].

## 5. Conclusions

To sum up, going beyond the pathophysiology and biomarkers mentioned above, preserving neuromuscular health through individualized rehabilitation programs, adequate nutrition and vitamin D supplementation may therefore protect not only mobility but also neurovascular and cardiometabolic integrity, especially in aging populations.

RT is consistently identified as a cornerstone of both prevention and treatment strategies for sarcopenia. Besides hypertrophy, physical activity improves mitochondrial renewal, glucose handling, and NMJ stability and stimulates the release of protective myokines. When aerobic and balance exercises are incorporated, additional gains occur in vascular regulation, coordination, and fall prevention. Moreover, the rapid development of tele-rehabilitation and digital exercise technologies has expanded therapeutic accessibility for older or mobility-restricted individuals. Future studies should address integrating standardized sarcopenia definitions, comparable outcome measures, and multimodal rehabilitation protocols to better delineate optimal exercise strategies and their multisystem effects across the aging continuum.

## Figures and Tables

**Figure 1 life-16-00068-f001:**
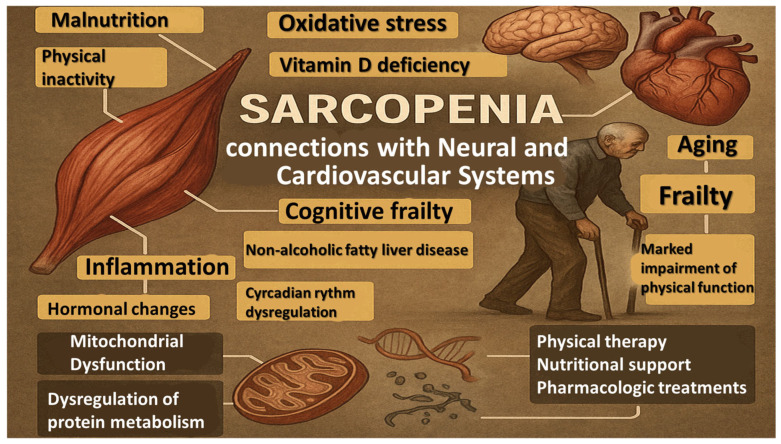
The figure synthesizes the extensive mechanistic content of Sarcopenia and clarifies how diverse biological, clinical, and lifestyle factors integrate into a single pathogenic continuum.

**Figure 2 life-16-00068-f002:**
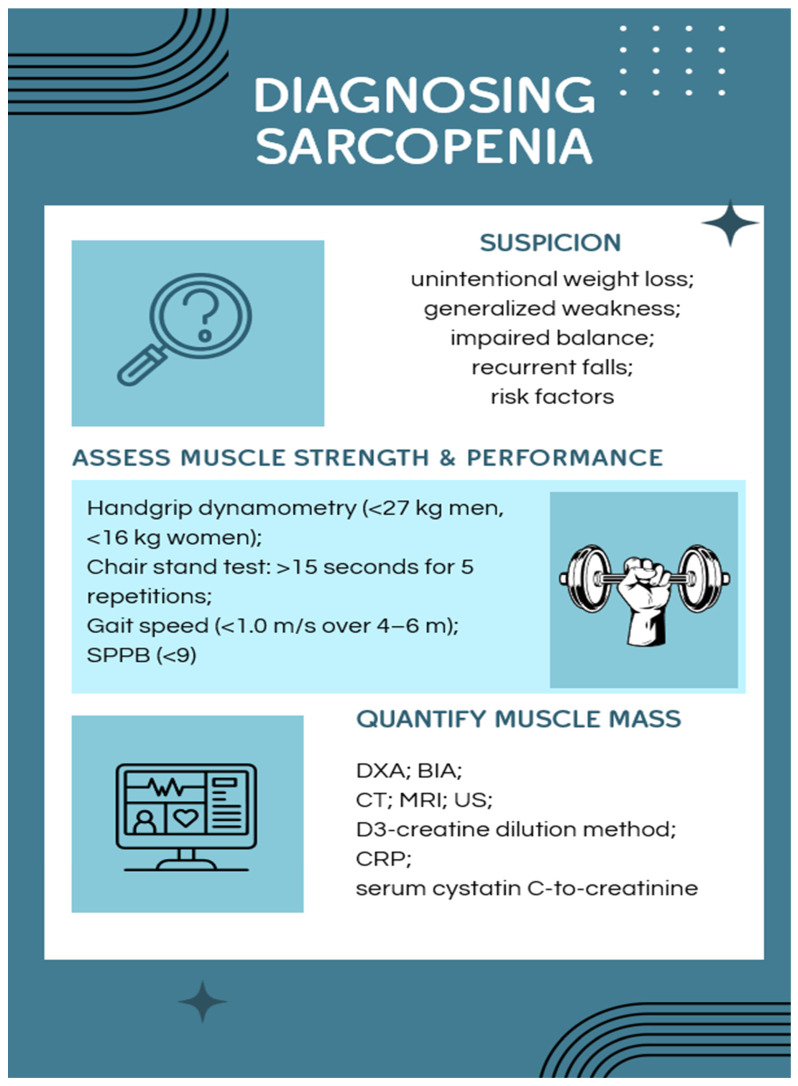
The figure provides a structured, stepwise diagnostic framework for identifying sarcopenia, aligned with contemporary clinical guidelines.; BIA—bioelectrical impedance analysis; CRP—C-reactive protein; CT—computed tomography; DXA—dual-energy X-ray absorptiometry; MRI—magnetic resonance imaging; SPPB—Short Physical Performance Battery; US—ultrasonography.

**Figure 3 life-16-00068-f003:**
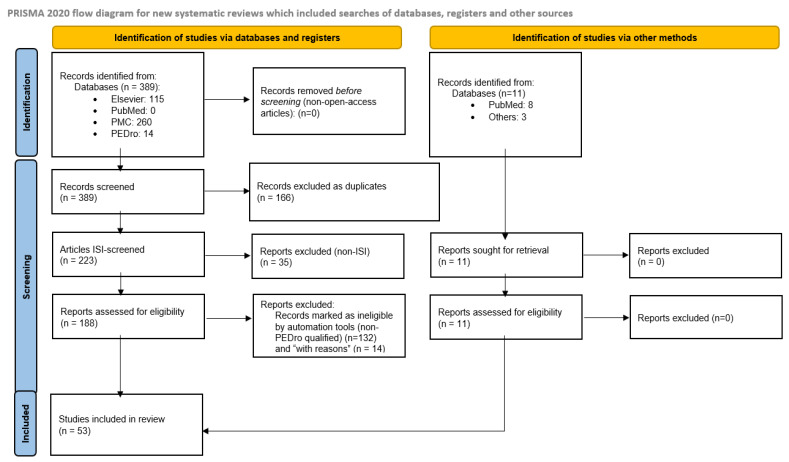
The figure represents the PRISMA 2020 flow diagram, illustrating the complete selection process for studies included in the systematic review. It visualizes every stage—from initial identification to final inclusion—ensuring methodological transparency and reproducibility.

**Figure 4 life-16-00068-f004:**
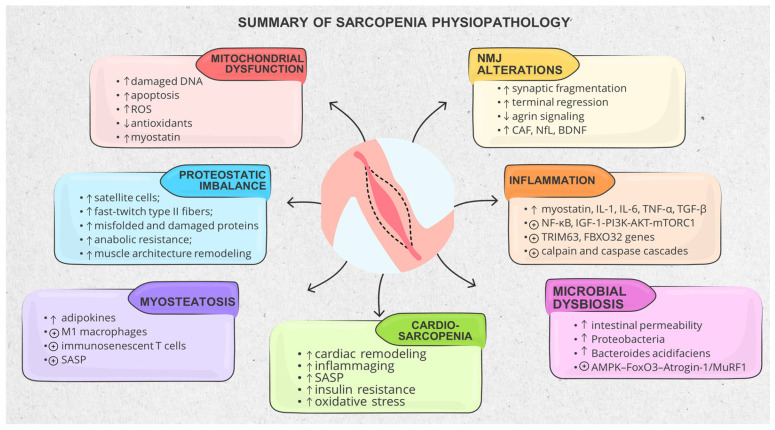
Biological mechanisms underlying sarcopenia—seven pathological clusters, each contributing to the progressive loss of skeletal muscle mass, strength, and function. BDNF—brain-derived neurotrophic factor; CAF—C-terminal agrin fragment; DNA—deoxyribonucleic acid; IGF-1-PI3K-AKT-mTORC1—insulin-like growth factor 1-phosphatidylinositol 3-kinase–serine/threonine kinase 1-mechanistic target of rapamycin complex 1; IL—interleukin; FBXO32—F-box protein 32 (Atrogin); MuRF1—Muscle Ring Finger 1; NF-κB—nuclear factor kappa B; NfL—neurofilament light chain; ROS—reactive oxygen species; SASP—senescence-associated secretory phenotype; TGF-β—transforming growth factor β; TNF-α—tumor necrosis factor-α; TRIM 63—Tripartite Motif 63; ⊕: activate; ↑: increase.

**Table 1 life-16-00068-t001:** Clinical implications of sarcopenia regarding mechanistic domains, assessment, and intervention; BIA—bioelectrical impedance analysis, DXA—dual-energy X-ray absorptiometry.

Mechanistic Domain	Clinical Relevance	Commonly Used Diagnostic Tools	Broad Intervention Domains
Mitochondrial dysfunction and metabolic stress	Reduced muscle endurance, early fatigability, impaired physical performance; contribution to multisystem vulnerability	Functional performance tests (e.g., handgrip strength, gait speed, chair stand test); imaging-based assessment of muscle quality	Resistance and aerobic exercise; nutritional optimization; correction of vitamin D deficiency; pharmacological agents targeting metabolic pathways (under investigation)
Proteostatic imbalance and anabolic resistance	Progressive loss of muscle mass and strength; reduced responsiveness to anabolic stimuli	DXA or BIA; functional performance tests	Progressive RT; individualized nutrition; hormonal or anabolic agents explored in selected contexts; multimodal rehabilitation
Inflammatory and catabolic signaling	Association with frailty, reduced physiological reserve, and comorbidity burden	Inflammatory markers (e.g., CRP) as contextual information; functional performance tests	Exercise-based rehabilitation; comorbidity management; lifestyle interventions; supportive anti-inflammatory or immunomodulatory approaches
Neuromuscular junction and motor unit degeneration	Impaired motor control, reduced coordination, increased fall risk	Gait speed, balance tests; neuro-functional performance tests	Task-oriented and balance training; neuromuscular activation exercises; experimental pharmacological modulation of neuromuscular signaling
Central nervous system involvement and muscle–brain crosstalk	Cognitive–motor interference, impaired adaptability to physical stressors	Cognitive screening tools; dual-task performance	Combined motor–cognitive training; multidisciplinary rehabilitation; centrally acting agents influencing neuroplasticity (context-dependent)
Cardio-sarcopenia (muscle–cardiovascular interactions)	Reduced exercise tolerance, poorer prognosis in cardiovascular disease, postoperative vulnerability	Functional capacity testing; body composition assessment; cardiovascular evaluation	Integrated cardiopulmonary and RT; cardiovascular rehabilitation; cardiometabolic drugs with potential muscle-related effects

## Data Availability

No new data were created or analyzed in this study. Data sharing is not applicable to this article.

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
