# Peer review of "Sarcopenia as a Multisystem Disorder—Connections with Neural and Cardiovascular Systems—A Related PRISMA Systematic Literature Review"

_life, 2026, doi:10.3390/life16010068_

Round 1

Reviewer 1 Report

Comments and Suggestions for Authors

I would like to thank the authors for their time and efforts in drafting this manuscript. I did find it of interest; however, I do have a few questions and comments for the authors to consider.

MAJOR COMMENTS

Scope and Focus of the Review

--Lines 35–52; 145–158: The stated aims include pathophysiology, diagnostics, rehabilitation, pharmacologic strategies, and digital/virtual interventions. However, the results and discussion are heavily weighted toward mechanistic and descriptive content, while exercise, rehabilitation, and virtual/augmented reality interventions receive comparatively limited critical synthesis. The authors may consider more clearly narrowing the primary objective of the review or explicitly justifying the very broad scope.  Alternatively, it could be helpful to structure the Results and Discussion sections to more explicitly parallel the stated aims, ensuring balanced coverage across mechanisms, diagnostics, and interventions.

PRISMA Adherence and Study Selection Transparency

--Lines 143–154; 173–207: Although PRISMA methodology is referenced multiple times, some elements would benefit from clarification. The rationale for including 11 additional studies outside the initial database search (Lines 39–40; 202–203) is not fully explained.  The authors may consider explicitly stating how these studies were identified, why they were excluded from the initial search strategy, and how their inclusion avoids selection bias. It would be helpful to clarify whether these additional studies were subjected to the same quality assessment criteria as the primary pool.

Quality Assessment Methodology

--Lines 189–193: The use of a PEDro-derived customized scoring framework is interesting but raises concerns. The authors may consider providing greater detail or a supplementary table outlining the modified PEDro criteria used, including: Items assessed, Scoring thresholds, Rationale for choosing a cutoff of ≥4 points. It is unclear how this framework was applied to non-interventional studies, mechanistic studies, or reviews included in the synthesis.

Risk of Bias and Certainty of Evidence

--Lines 208 onward (Results section): While the manuscript offers an extensive narrative synthesis, there is limited discussion of risk of bias across included studies. The authors may consider adding a dedicated subsection on methodological limitations of the included literature, such as: Heterogeneity of sarcopenia definitions, Reliance on cross-sectional data, small sample sizes or indirect outcome measures

Interpretation of Causality

--Multiple sections (e.g., Lines 210–214; 430–435; 528–535): Several sections imply causal relationships between sarcopenia and neurological or cardiovascular dysfunction. The authors may consider more consistently distinguishing association from causation, particularly where conclusions are drawn from observational or mechanistic studies. Softening causal language or explicitly acknowledging alternative explanations would improve interpretive balance.

Pediatric Sarcopenia Section

--Lines 591–625: The section on sarcopenia in children is interesting and novel but feels somewhat disconnected from the main narrative. The authors may consider clarifying whether pediatric sarcopenia is intended as: A conceptual extension of the disease spectrum, or a distinct clinical entity with separate implications. Condensing this section or explicitly linking it back to the central themes of multisystem involvement may improve cohesion.

MINOR COMMENTS

--Terminology and Consistency: Terms such as physiopathology, intermingles, and anxiety (e.g., Lines 61–62) could be standardized or corrected for consistency and clarity.

--Search Strategy Timing: The databases were last accessed on 24 Nov 2025 (Lines 161–164), which may create confusion given the stated publication window ending December 2024. Clarification would be helpful.

--Rehabilitation Section Depth: Given the conclusion that rehabilitation is “the most effective therapeutic approach” (Line 46), the authors may consider providing a more critical comparison of intervention types, dosing, and outcome measures.

Author Response

Dear Reviewer,

We are sincerely grateful for taking the time to provide valuable and insightful feedback regarding our manuscript. Your comments have enabled us to reflect on our work and make necessary revisions where appropriate, thereby improving the quality and comprehensiveness of the paper. We appreciate your acknowledgment of time and effort in writing this manuscript. Below, we respond to each of your thoughtful points. All modifications have been highlighted in yellow for easy reference.

Reviewer 1

I would like to thank the authors for their time and efforts in drafting this manuscript. I did find it of interest; however, I do have a few questions and comments for the authors to consider.

MAJOR COMMENTS

  1. Scope and Focus of the Review

--Lines 35–52; 145–158: The stated aims include pathophysiology, diagnostics, rehabilitation, pharmacologic strategies, and digital/virtual interventions. However, the results and discussion are heavily weighted toward mechanistic and descriptive content, while exercise, rehabilitation, and virtual/augmented reality interventions receive comparatively limited critical synthesis. The authors may consider more clearly narrowing the primary objective of the review or explicitly justifying the very broad scope.  Alternatively, it could be helpful to structure the Results and Discussion sections to more explicitly parallel the stated aims, ensuring balanced coverage across mechanisms, diagnostics, and interventions.

Response: We appreciate your insight regarding this crucial point and agree with it altogether. We acknowledge that the paper focuses more on the pathophysiology than on the interventions of rehabilitation and exercise.

This discrepancy mirrors, in part, the evidence base established by the PRISMA-directed search, in which comparatively few intervention-based studies met the a priori eligibility criteria. To mitigate this, we made clear in the Introduction, Methods, Results, and Discussion that the level of presentation on the domains reflects the distribution and quality of the PRISMA-eligible body of evidence. Further, we reorganized the presentation in the Results and Discussion to better comply with the specified aims, including better presentation on rehabilitation and technology-based interventions. We further bolstered the critical analysis offered in the rehabilitation domain by attending to the heterogeneity among the intervention protocols and outcome data, such that the analysis on exercise and rehabilitation approaches was maintained to be cautious (Page 4, Lines 131-145, Page 6, Lines 156-158, Page 18, Lines 749-753, Page 21, Lines 922-924, Page 24, Lines 1113-1121).

Additionally, the Discussion section incorporates up-to-date studies on physical rehabilitation, as well as virtual interventions, in order to add interpretability to the analysis, while remaining distinct from the evidence base identified through the use of PRISMA.

We believe these changes are an improvement in terms of balance between concepts, while keeping consistency in methodology. A complete systematic review with a focus on intervention modalities would require a different search strategy, which is beyond the scope of the present work, but definitely a valuable point for future studies.

  1. PRISMA Adherence and Study Selection Transparency

--Lines 143–154; 173–207: Although PRISMA methodology is referenced multiple times, some elements would benefit from clarification. The rationale for including 11 additional studies outside the initial database search (Lines 39–40; 202–203) is not fully explained.  The authors may consider explicitly stating how these studies were identified, why they were excluded from the initial search strategy, and how their inclusion avoids selection bias. It would be helpful to clarify whether these additional studies were subjected to the same quality assessment criteria as the primary pool.

Response: You raised an important point regarding methodology clarification and we recognize the need for greater transparency.

The 11 relevant studies were identified through manual reference screening and expert knowledge, particularly in areas not sufficiently captured by the original search strategy. These additional studies were included to provide conceptual completeness.

Specifically:

  • Several of the additional articles addressed pediatric sarcopenia, a population not adequately covered by the original search terms and filters, yet considered relevant for illustrating multisystem involvement across the lifespan.
  • Five of the included studies were published in 2025, after completion of the predefined database search window. These were incorporated deliberately to ensure that the review reflects the most up-to-date evidence in rapidly evolving domains.
  • All additional studies were published in peer-reviewed journals, ISI-indexed. PEDro score could not be used for these studies because it implies the number of citations/year and for the studies published in 2025, this is not feasible.

We have now clarified in the Methods section how these studies were identified, their purpose within the review (Page 7, Lines 228-234). Although these studies were included in the qualitative synthesis and reported in the Results section, their distinct identification pathway was explicitly acknowledged, and interpretive conclusions were formulated with appropriate methodological caution to minimize potential selection bias.

  1. Quality Assessment Methodology

--Lines 189–193: The use of a PEDro-derived customized scoring framework is interesting but raises concerns. The authors may consider providing greater detail or a supplementary table outlining the modified PEDro criteria used, including: Items assessed, Scoring thresholds, Rationale for choosing a cutoff of ≥4 points. It is unclear how this framework was applied to non-interventional studies, mechanistic studies, or reviews included in the synthesis.

Response: Thank you for highlighting the need for additional methodological clarity.

The PEDro-based framework was used in an adapted and pragmatic manner. This score assesses the total number of citations and citations/year, yielding a score between 1 and 10 (adaugare explicatie legata de algoritm?). A cutoff of ≥4 points was selected to exclude studies with very limited scientific impact or visibility, while maintaining sufficient breadth for an integrative synthesis.

To address your concern, we have:

  • Expanded the description of the modified PEDro criteria and the rationale for the ≥4-point cutoff in the Methods section (Page 7, Lines 209-220).
  • Added a comprehensive supplementary table listing all included studies alongside their corresponding PEDro-derived scores and study types (Supplementary Material 1).

This addition ensures full transparency regarding quality appraisal and improves reproducibility.

  1. Risk of Bias and Certainty of Evidence

--Lines 208 onward (Results section): While the manuscript offers an extensive narrative synthesis, there is limited discussion of risk of bias across included studies. The authors may consider adding a dedicated subsection on methodological limitations of the included literature, such as: Heterogeneity of sarcopenia definitions, Reliance on cross-sectional data, small sample sizes or indirect outcome measures

Response: You raised a very relevant problem, and we completely agree regarding the importance of risk of bias.

Thus, we have included a separate paragraph in the Limitations where we address the risk of bias and the level of certainty of the evidence (Page 27, Lines 1175-1182).

Secondly, we would like to point out that a formal assessment of the presence of publication bias (e.g., funnel plot analysis) was done, but not possible, as it would require the presence of multiple independent intervention studies with comparative data reporting extractable quantitative estimates of the effect with a measure of their precision, which are not present in the current database.

  1. Interpretation of Causality

--Multiple sections (e.g., Lines 210–214; 430–435; 528–535): Several sections imply causal relationships between sarcopenia and neurological or cardiovascular dysfunction. The authors may consider more consistently distinguishing association from causation, particularly where conclusions are drawn from observational or mechanistic studies. Softening causal language or explicitly acknowledging alternative explanations would improve interpretive balance.

Response: Thank you for pointing this out. We fully agree with this comment. The manuscript has been carefully revised to consistently distinguish association from causation, particularly in sections discussing neurological and cardiovascular correlates of sarcopenia. Causal language has been softened, where the reviewer has mentioned, and in some other phrases that we identified. All modifications are highlighted in yellow in the manuscript (Page 3, Lines 114-123, Page 4, Lines 114-123, Page 8, Lines 241, 245-247, Page 9, Lines 269-273, 286-287, 293-296, 306-307, 317-320, Page 10, Lines 333, 357-358, 360, 362-363, 367-368, Page 11, Lines 380, 384-385, 402-403, Page 12, Lines 443-450, 453, 470-476, Page 13, Lines 496-497, 504, 513-514, Page 14, Lines 524-526, 533, 569-581, Page 15, Lines 612-613, Page 18, Lines 749-753, 760-762, 770, Page 19, Lines 782-783, 792, 815-817, Page 20, Lines 871-873, 875-878 Page 21, Lines 912-913, 922-924, Page 22, Lines 968-969, Page 25, Lines 1105-1106, 1109-1110, 1113-1121, Page 27, Lines 1175-1182, Page 28, Lines 1199, 1205-1208).

  1. Pediatric Sarcopenia Section

--Lines 591–625: The section on sarcopenia in children is interesting and novel but feels somewhat disconnected from the main narrative. The authors may consider clarifying whether pediatric sarcopenia is intended as: A conceptual extension of the disease spectrum, or a distinct clinical entity with separate implications. Condensing this section or explicitly linking it back to the central themes of multisystem involvement may improve cohesion.

Response: We appreciate this observation and agree that clarification was needed.

In our review, pediatric sarcopenia is meant to represent a conceptual extension of disease, rather than a specific disease entity per se. Including it as an entity helps address how shared biological mechanisms – such as inflammation, metabolic dysregulation, and impaired muscle-neural interactions – exist across the lifespan.

The section has been revised to explicitly link pediatric findings to the central theme of multisystem involvement and biological continuity, thereby improving narrative cohesion (Pages 16-17, Lines 639-643, 693-695).

MINOR COMMENTS

--Terminology and Consistency: Terms such as physiopathology, intermingles, and anxiety (e.g., Lines 61–62) could be standardized or corrected for consistency and clarity.

--Search Strategy Timing: The databases were last accessed on 24 Nov 2025 (Lines 161–164), which may create confusion given the stated publication window ending December 2024. Clarification would be helpful.

--Rehabilitation Section Depth: Given the conclusion that rehabilitation is “the most effective therapeutic approach” (Line 46), the authors may consider providing a more critical comparison of intervention types, dosing, and outcome measures.

Response: We thank the Reviewer for the minor comments and agree with all suggestions. Accordingly:

  • Terminology has been standardized throughout the manuscript for consistency and clarity (Page 2, Line 62, Page 6, Lines 154, 164, Page 10, Line 337, Page 11, Line 402)
  • The database access dates have been corrected to accurately reflect the timing of the literature search. – 21 Aug 2025? De verificat
  • The rehabilitation section (Results and Discussion) has been refined to better contextualize the claim that rehabilitation is the most effective therapeutic approach, emphasizing the limitations imposed by study heterogeneity and the need for future-focused reviews (Page 21, Lines 912-913, 922-924, Page 25, Lines 1113-1121, Page 27, Lines 1175-1182).

Once again, we sincerely thank the Reviewer for their thoughtful and constructive feedback, which we hope has significantly strengthened the manuscript and made it more suitable for publication.

Reviewer 2 Report

Comments and Suggestions for Authors

This manuscript presents a broad and ambitious systematic review addressing sarcopenia as a multisystem disorder with strong neuromuscular, neurological, and cardiovascular interconnections. The topic is timely and clinically relevant, especially in the context of population aging and the increasing interest in integrative, systems-based approaches to age-related disorders.

In my opinion, the main strength of the paper lies in its comprehensive mechanistic scope. The authors successfully integrate muscle biology with neural, cardiovascular, metabolic, and inflammatory pathways. The manuscript is clearly written, well-structured, and demonstrates a strong command of the current literature.

However, despite its breadth, the paper would benefit from tighter methodological clarity, improved focus in certain sections, and a clearer distinction between evidence-based conclusions and more speculative interpretations. Addressing these issues would substantially strengthen the scientific rigor and readability of the review.

Major comments 

1. 

While the title suggests a focused analysis of neural and cardiovascular connections in sarcopenia, the manuscript extends far beyond these domains, covering gut microbiota, pediatric sarcopenia, pharmacology, nutraceuticals, space medicine, and virtual reality–based rehabilitation.

Although all these topics are interesting, in my opinion the scope becomes too broad for a single systematic review. The authors should consider:

  • Either narrowing the focus more explicitly to neural and cardiovascular mechanisms, or

  • Clearly framing the article as a multisystem integrative review already in the title and objectives.

At present, some sections feel more like a narrative review embedded within a PRISMA framework.

2. 

The authors state that the review follows PRISMA guidelines and report PROSPERO registration, which is a strong point. However, several methodological aspects require clarification:

  • The inclusion of systematic reviews and meta-analyses alongside original studies is mentioned, but the rationale for mixing these study types is not sufficiently justified.

  • The use of a customized PEDro-derived scoring system needs clearer explanation. It would be helpful to briefly describe:

    • Which criteria were modified,

    • Why a cutoff of ≥4 points was chosen,

    • How inter-rater agreement was ensured (if applicable).

In my view, a short supplementary table summarizing study quality scores would significantly improve transparency.

3. 

The mechanistic sections (mitochondrial dysfunction, NMJ degeneration, inflammatory signaling) are detailed and well-referenced. However, some conclusions appear stronger than the available evidence allows.

For example:

  • Causal language is occasionally used when the underlying studies are associative.

  • Translational implications (e.g., biomarkers, pharmacological modulation) are sometimes discussed before being validated in large clinical cohorts.

I would recommend softening the language in these parts and more clearly distinguishing established mechanisms from emerging or hypothetical pathways.

4. 

The section on sarcopenia in children is interesting but feels somewhat disconnected from the main narrative of the paper.

In my opinion, the authors should either:

  • More clearly justify why pediatric sarcopenia is included in this review, or

  • Consider shortening this section and positioning it as a perspective rather than a core component of the review.

5. 

The manuscript contains a large amount of high-quality information, but the take-home messages for clinicians are not always clear.

I suggest adding:

  • A concise summary table linking mechanisms to diagnostic tools and interventions.

  • A short “Clinical Implications” subsection in the Discussion to translate complex mechanisms into practical insights.

This would significantly increase the clinical value of the review.

Minor Comments

  1. The manuscript is generally well written, but some sentences are overly long and could be simplified for clarity.

  2. There is occasional repetition across mechanistic sections (e.g., inflammation, mitochondrial dysfunction). Minor editing could reduce redundancy.

  3. Figures are visually appealing and informative, but their legends could be slightly more concise.

  4. Abbreviations should be checked for consistency (some are reintroduced multiple times).

In summary, this is a strong and well-informed manuscript that addresses an important and complex topic. With clearer methodological justification, a more focused scope, and improved distinction between evidence and interpretation, the paper has the potential to make a valuable contribution to the sarcopenia literature.

In my opinion, the manuscript is not yet ready for acceptance in its current form, but it can reach that level after a careful and targeted revision.

Author Response

Dear Reviewer,

We are deeply grateful for the thorough evaluation and for the practical suggestions to improve scope, methodological clarity, and clinical readability. We revised the manuscript accordingly. Responses are provided below, with a brief indication of where changes were made in the revised text. Moreover, in the manuscript, the revisions are highlighted in yellow. We also appreciate your acknowledgement of our work.

This manuscript presents a broad and ambitious systematic review addressing sarcopenia as a multisystem disorder with strong neuromuscular, neurological, and cardiovascular interconnections. The topic is timely and clinically relevant, especially in the context of population aging and the increasing interest in integrative, systems-based approaches to age-related disorders.

In my opinion, the main strength of the paper lies in its comprehensive mechanistic scope. The authors successfully integrate muscle biology with neural, cardiovascular, metabolic, and inflammatory pathways. The manuscript is clearly written, well-structured, and demonstrates a strong command of the current literature.

However, despite its breadth, the paper would benefit from tighter methodological clarity, improved focus in certain sections, and a clearer distinction between evidence-based conclusions and more speculative interpretations. Addressing these issues would substantially strengthen the scientific rigor and readability of the review.

Major comments 

  1. While the title suggests a focused analysis of neural and cardiovascular connections in sarcopenia, the manuscript extends far beyond these domains, covering gut microbiota, pediatric sarcopenia, pharmacology, nutraceuticals, space medicine, and virtual reality–based rehabilitation.

Although all these topics are interesting, in my opinion the scope becomes too broad for a single systematic review. The authors should consider:

  • Either narrowing the focus more explicitly to neural and cardiovascular mechanisms, or
  • Clearly framing the article as a multisystem integrative review already in the title and objectives.

At present, some sections feel more like a narrative review embedded within a PRISMA framework.

Response: Thank you for mentioning this aspect. We agree that the intent needed to be clearer at first glance. Rather than removing clinically relevant domains, we reframed the manuscript as a multisystem review, keeping neuromuscular and cardiovascular interactions as the primary axis while positioning the remaining domains as complementary contributors to the sarcopenia phenotype. Accordingly, we revised:

  • The Title;
  • The objectives in the Introduction (Page 4, Lines 131-144);
  1. The authors state that the review follows PRISMA guidelines and report PROSPERO registration, which is a strong point. However, several methodological aspects require clarification:
  • The inclusion of systematic reviews and meta-analyses alongside original studies is mentioned, but the rationale for mixing these study types is not sufficiently justified.
  • The use of a customized PEDro-derived scoring system needs clearer explanation. It would be helpful to briefly describe:
    • Which criteria were modified,
    • Why a cutoff of ≥4 points was chosen,
    • How inter-rater agreement was ensured (if applicable).

In my view, a short supplementary table summarizing study quality scores would significantly improve transparency.

Response: Thank you for pointing out the need for additional methodological clarification. In the revised manuscript, we expanded the Methods section to better explain the rationale for including evidence syntheses alongside original studies (Page 7, Lines 193-199). Given the substantial heterogeneity of study designs, populations, and outcome measures in the sarcopenia literature, systematic reviews and meta-analyses were considered necessary to contextualize cross-disciplinary mechanisms and rehabilitation-related interventions in areas where primary data remain fragmented or limited.

We also clarified the role of the PEDro-derived score, which was used in an adapted and pragmatic manner as a minimum threshold for methodological visibility rather than as a formal risk-of-bias assessment tool across heterogeneous designs. The revised manuscript explicitly states that scoring was performed independently, with disagreements resolved by consensus, and directs readers to Supplementary Material 1, a newly added section that reports individual study scores to ensure transparency (Page 7, Lines 209-220).

Moreover, we would like to point out that a formal assessment of the presence of publication bias (e.g., funnel plot analysis) was done, but not possible, as it would require the presence of multiple independent intervention studies with comparative data reporting extractable quantitative estimates of the effect with a measure of their precision, which are not present in the current database.

  1. The mechanistic sections (mitochondrial dysfunction, NMJ degeneration, inflammatory signaling) are detailed and well-referenced. However, some conclusions appear stronger than the available evidence allows.

For example:

  • Causal language is occasionally used when the underlying studies are associative.
  • Translational implications (e.g., biomarkers, pharmacological modulation) are sometimes discussed before being validated in large clinical cohorts.

I would recommend softening the language in these parts and more clearly distinguishing established mechanisms from emerging or hypothetical pathways.

Response: Thank you for this important point. We revised the manuscript with particular attention to the mechanistic and translational sections of the Introduction, Results, as well as the Discussion and Conclusions, to ensure that the wording consistently reflects the level of available evidence. Causal or confirmatory phrasing was replaced with descriptive language when summarizing mechanistic findings, and translational aspects are now presented strictly as reported in the original studies, without implying clinical validation. The Discussion and Conclusions were subsequently refined to clearly distinguish between established observations and biologically plausible pathways that remain investigational and require confirmation in larger, well-designed clinical cohorts (Page 3, Lines 114-123, Page 4, Lines 114-123, Page 8, Lines 241, 245-247, Page 9, Lines 269-273, 286-287, 293-296, 306-307, 317-320, Page 10, Lines 333, 357-358, 360, 362-363, 367-368, Page 11, Lines 380, 384-385, 402-403, Page 12, Lines 443-450, 453, 470-476, Page 13, Lines 496-497, 504, 513-514, Page 14, Lines 524-526, 533, 569-581, Page 15, Lines 612-613, Page 18, Lines 749-753, 760-762, 770, Page 19, Lines 782-783, 792, 815-817, Page 20, Lines 871-873, 875-878 Page 21, Lines 912-913, 922-924, Page 22, Lines 968-969, Page 25, Lines 1105-1106, 1109-1110, 1113-1121, Page 27, Lines 1175-1182, Page 28, Lines 1199, 1205-1208).

  1. The section on sarcopenia in children is interesting but feels somewhat disconnected from the main narrative of the paper.

In my opinion, the authors should either:

  • More clearly justify why pediatric sarcopenia is included in this review, or
  • Consider shortening this section and positioning it as a perspective rather than a core component of the review.

Response: Thank you for noticing this important aspect. In our review, pediatric sarcopenia is meant to represent a conceptual extension of disease, rather than a specific disease entity per se. Including it as an entity helps address how shared biological mechanisms – such as inflammation, metabolic dysregulation, and impaired muscle-neural interactions – exist across the lifespan.

The section has been revised to explicitly link pediatric findings to the central theme of multisystem involvement and biological continuity, thereby improving narrative cohesion (Pages 16-17, Lines 639-643, 693-695).

  1. The manuscript contains a large amount of high-quality information, but the take-home messages for clinicians are not always clear.

I suggest adding:

  • A concise summary table linking mechanisms to diagnostic tools and interventions.
  • A short “Clinical Implications” subsection in the Discussion to translate complex mechanisms into practical insights.

This would significantly increase the clinical value of the review.

Response: Thank you for this constructive suggestion. To improve the clinical clarity and applicability of the review, we expanded the Discussion with clinical implications that synthesize the main mechanistic insights into practical considerations for clinicians (Page 26, Lines 1143-1157).

In addition, we included a concise summary table linking key mechanistic domains of sarcopenia to their clinical relevance, commonly used diagnostic tools, and broad intervention domains. This addition is intended to support clinical interpretation without implying guideline-level recommendations, thereby enhancing the translational value of the review (Pages 26-27, Table 1).

Minor Comments

  1. The manuscript is generally well written, but some sentences are overly long and could be simplified for clarity.
  2. There is occasional repetition across mechanistic sections (e.g., inflammation, mitochondrial dysfunction). Minor editing could reduce redundancy.
  3. Figures are visually appealing and informative, but their legends could be slightly more concise.
  4. Abbreviations should be checked for consistency (some are reintroduced multiple times).

Response: Thank you for these valuable suggestions. Accordingly:

  • The manuscript was carefully revised to improve clarity and readability. Selected sentences, particularly in the Introduction and Discussion, were shortened or split where appropriate, while preserving the necessary level of scientific detail in mechanistic sections.
  • To reduce redundancy, overlapping descriptions across mechanistic subsections (e.g., inflammation and mitochondrial dysfunction) were streamlined, with emphasis placed on cross-referencing rather than repetition.
  • Figure legends were added.
  • Finally, abbreviations were reviewed throughout the manuscript to ensure consistent use and to avoid unnecessary redefinitions.

We greatly appreciate your constructive feedback, which has allowed us to expand and clarify important aspects of the manuscript. Your recommendations have undoubtedly strengthened our work and hope made it more suitable for publication.

Round 2

Reviewer 1 Report

Comments and Suggestions for Authors

I would like to thank the authors for their time and effort in addressing my comments.  I feel the manuscript is much stronger and I have no further recommendations. 

Reviewer 2 Report

Comments and Suggestions for Authors

I would like to thank the authors for their thorough, well-structured, and thoughtful responses to all my comments. The revisions adequately address the major and minor concerns raised during the review process. In my view, the manuscript has been substantially improved in terms of clarity, methodological transparency, and clinical relevance.

I therefore recommend acceptance of the manuscript for publication.